# Downregulation of glial genes involved in synaptic function mitigates Huntington's disease pathogenesis

Tarik Seref Onur[1,2,3†], Andrew Laitman[2,4,5†], He Zhao[2], Ryan Keyho[2], Hyemin Kim[2], Jennifer Wang[2], Megan Mair[1,2,3], Huilan Wang[6], Lifang Li[1,2], Alma Perez[2], Maria de Haro[1,2], Ying-Wooi Wan[2], Genevera Allen[2,7], Boxun Lu[6], Ismael Al-Ramahi[1,2], Zhandong Liu[2,4,5], Juan Botas[1,2,3,4]*

[1]Department of Molecular and Human Genetics, Baylor College of Medicine, Houston, United States; [2]Jan and Dan Duncan Neurological Research Institute at Texas Children's Hospital, Houston, United States; [3]Genetics & Genomics Graduate Program, Baylor College of Medicine, Houston, United States; [4]Quantitative & Computational Biosciences, Baylor College of Medicine, Houston, United States; [5]Department of Pediatrics, Baylor College of Medicine, Houston, United States; [6]State Key Laboratory of Medical Neurobiology and MOE Frontiers Center for Brain Science, Fudan University, Shanghai, China; [7]Departments of Electrical & Computer Engineering, Statistics and Computer Science, Rice University, Houston, United States

**Abstract** Most research on neurodegenerative diseases has focused on neurons, yet glia help form and maintain the synapses whose loss is so prominent in these conditions. To investigate the contributions of glia to Huntington's disease (HD), we profiled the gene expression alterations of *Drosophila* expressing human mutant *Huntingtin* (m*HTT*) in either glia or neurons and compared these changes to what is observed in HD human and HD mice striata. A large portion of conserved genes are concordantly dysregulated across the three species; we tested these genes in a high-throughput behavioral assay and found that downregulation of genes involved in synapse assembly mitigated pathogenesis and behavioral deficits. To our surprise, reducing d*NRXN3* function in glia was sufficient to improve the phenotype of flies expressing m*HTT* in neurons, suggesting that mHTT's toxic effects in glia ramify throughout the brain. This supports a model in which dampening synaptic function is protective because it attenuates the excitotoxicity that characterizes HD.

*For correspondence:
jbotas@bcm.edu

†These authors contributed equally to this work

Competing interests: The authors declare that no competing interests exist.

## Introduction

Neurodegenerative conditions involve a complex cascade of events that takes many years to unfold. Even in the case of inherited disorders due to mutation in a single gene, such as Huntington's disease (HD), the downstream ramifications at the molecular level are astonishingly broad. Caused by a CAG repeat expansion in *Huntingtin* (*HTT*) (***The Huntington's Disease Collaborative Research Group, 1993***), HD pathology is prominent in the striatum and cortex, yet transcriptomic studies consistently reveal thousands of changes in gene expression across the brain and different neuronal cell types, involving pathways ranging from autophagy to vesicular trafficking (***Saudou and Humbert, 2016***). To disentangle changes that are pathogenic from those that represent the brain's effort to compensate for the disease, we recently integrated transcriptomics with in silico analysis and high-throughput in vivo screening using a *Drosophila* model of HD (***Al-Ramahi et al., 2018***). This study demonstrated that HD pathogenesis is driven by upregulation of genes involved in the actin

**eLife digest** When a neuron dies, through injury or disease, the body loses all communication that passes through it. The brain compensates by rerouting the flow of information through other neurons in the network. Eventually, if the loss of neurons becomes too great, compensation becomes impossible. This process happens in Alzheimer's, Parkinson's, and Huntington's disease. In the case of Huntington's disease, the cause is mutation to a single gene known as huntingtin. The mutation is present in every cell in the body but causes particular damage to parts of the brain involved in mood, thinking and movement.

Neurons and other cells respond to mutations in the huntingtin gene by turning the activities of other genes up or down, but it is not clear whether all of these changes contribute to the damage seen in Huntington's disease. In fact, it is possible that some of the changes are a result of the brain trying to protect itself. So far, most research on this subject has focused on neurons because the huntingtin gene plays a role in maintaining healthy neuronal connections. But, given that all cells carry the mutated gene, it is likely that other cells are also involved. The glia are a diverse group of cells that support the brain, providing care and sustenance to neurons. These cells have a known role in maintaining the connections between neurons and may also have play a role in either causing or correcting the damage seen in Huntington's disease.

The aim of Onur et al. was to find out which genes are affected by having a mutant huntingtin gene in neurons or glia, and whether severity of Huntington's disease improved or worsened when the activity of these genes changed. First, Onur et al. identified genes affected by mutant huntingtin by comparing healthy human brains to the brains of people with Huntington's disease. Repeating the same comparison in mice and fruit flies identified genes affected in the same way across all three species, revealing that, in Huntington's disease, the brain dials down glial cell genes involved in maintaining neuronal connections.

To find out how these changes in gene activity affect disease severity and progression, Onur et al. manipulated the activity of each of the genes they had identified in fruit flies that carried mutant versions of huntingtin either in neurons, in glial cells or in both cell types. They then filmed the flies to see the effects of the manipulation on movement behaviors, which are affected by Huntington's disease. This revealed that purposely lowering the activity of the glial genes involved in maintaining connections between neurons improved the symptoms of the disease, but only in flies who had mutant huntingtin in their glial cells. This indicates that the drop in activity of these genes observed in Huntington's disease is the brain trying to protect itself.

This work suggests that it is important to include glial cells in studies of neurological disorders. It also highlights the fact that changes in gene expression as a result of a disease are not always bad. Many alterations are compensatory, and try to either make up for or protect cells affected by the disease. Therefore, it may be important to consider whether drugs designed to treat a condition by changing levels of gene activity might undo some of the body's natural protection. Working out which changes drive disease and which changes are protective will be essential for designing effective treatments.

cytoskeleton and inflammation, but that neurons compensate by downregulating the expression of genes involved in synaptic biology and calcium signaling.

The finding that synaptic changes were protective caught our attention because HTT itself is necessary for normal synaptogenesis and maintenance within the cortico-striatal circuit (*McKinstry et al., 2014*), largely through its role in retrograde axonal trafficking of neurotrophic factors (*Saudou and Humbert, 2016*). But synapses involve more than just neurons: glial cells also contribute to synapse formation, function, and elimination (*Filipello et al., 2018*; *McKinstry et al., 2014*; *Octeau et al., 2018*; *Stogsdill et al., 2017*). There is, in fact, emerging evidence that various glial subtypes affect outcomes in HD. The accumulation of mutant Huntingtin(mHTT) in astrocytes and oligodendrocytes hinders their development and function and contributes to disease pathophysiology (*Benraiss et al., 2016*; *Ferrari Bardile et al., 2019*; *Osipovitch et al., 2019*; *Wood et al., 2018*). Conversely, healthy glia can improve the disease phenotype in HD mice (*Benraiss et al., 2016*). Recent studies using single-cell sequencing in astrocytes isolated from post-mortem tissue

from HD patients and mouse models of HD (*Al-Dalahmah et al., 2020*; *Diaz-Castro et al., 2019*) developed molecular profiles that distinguish HD-affected astrocytes from astrocytes found in healthy brain tissue, but the physiological consequences of the gene expression changes were unclear. Whether mHTT affects glial participation in synapse formation or maintenance remains unknown, but then, we are only just now beginning to understand the range of glial types and their functions (*Bayraktar et al., 2020*; *Darmanis et al., 2015*).

The combination of synaptic degeneration in HD and the fact that both HTT and glia contribute to synaptic formation and maintenance led us to further investigate the influence of *mHTT* in glia. Because *Drosophila* have been used to elucidate glial biology (*Freeman and Doherty, 2006*; *Olsen and Feany, 2019*; *Pearce et al., 2015*; *Ziegenfuss et al., 2012*) and are a tractable model system for studying HD and other neurodegenerative diseases (*Al-Ramahi et al., 2018*; *Bondar et al., 2018*; *Donnelly et al., 2020*; *Fernandez-Funez et al., 2000*; *Filimonenko et al., 2010*; *Goodman et al., 2019*; *Ochaba et al., 2014*; *Olsen and Feany, 2019*; *O'Rourke et al., 2013*; *Rousseaux et al., 2018*; *Yuva-Aydemir et al., 2018*), we decided to generate flies that express m*HTT* solely in glia so that we could compare their transcriptomic signature with that of flies expressing m*HTT* in neurons. We took an unbiased approach, first establishing the repertoire of evolutionarily conserved genes that show concordant expression changes across HD human and mouse striata and HD fly brains. We then integrated this comparative transcriptomic data with high-throughput in vivo behavioral screening to acquire insight into glial contributions to HD pathogenesis and identify disease-modifying targets that mitigate the HD phenotype.

## Results

### The HD transcriptome is conserved among evolutionarily distant model systems

To study the contributions of neurons and glia to HD pathogenesis, we first needed to define a transcriptomic signature that would enable us to move across species (human, mouse, and fly) (*Figure 1A*). We began with human tissue. Since the striatum is the brain region most prominently affected in HD, we compared the gene expression profiles of human post-mortem striatal samples from healthy individuals and patients with HD, from different stages of the disease (i.e., Vonsattel Grade 0–4) (*Hodges et al., 2006*; *Vonsattel et al., 1985*). We identified 1852 downregulated and 1941 upregulated differentially expressed genes (DEGs) in patients with HD compared to healthy individuals (*Figure 1B*).

We then reanalyzed published RNA-seq data from mouse striata using an allelic series of knock-in mouse models with varying CAG repeat lengths at 6 months of age (*Langfelder et al., 2016*). Because it is unclear which CAG tract length in mice most faithfully recapitulates HD pathogenesis, the triplet repeat length was treated as a continuous trait, and we narrowed our analysis to DEGs that correlate with increasing CAG repeat length. Comparing the striata of wildtype mice to the knock-in HD mouse models, there were 3575 downregulated and 3634 upregulated DEGs (*Figure 1B*). (The greater genome coverage provided by RNA-seq [*Miller et al., 2014*] yielded larger datasets for mouse and, below, for *Drosophila* than for humans.)

We performed RNA-seq leveraging *Drosophila* HD models (*Kaltenbach et al., 2007*; *Romero et al., 2008*) (see Materials and methods) to compare the effect of expressing m*HTT* in either neurons or glia. The binary GAL4-*UAS* system was used to drive the expression of human m*HTT* either in neurons (*elav >GAL4)* or glia (*repo >GAL4*). Both full-length (HTT$^{FLQ200}$) and N-terminal (HTT$^{NT231Q128}$) models were used in this set of experiments since both the full protein and N-terminal HD fragments accumulate in the human brain as a result of proteolysis and mis-splicing (*Kim et al., 2001*; *Neueder et al., 2017*; *Sathasivam et al., 2013*; *Wellington et al., 2002*). Principal component analysis (PCA) showed that the greatest differences between samples are attributable to the cell-specific drivers, and not to the use of N-terminal versus full-length protein (*Figure 1—figure supplement 1*). Expressing m*HTT* in neurons resulted in 3058 downregulated and 2979 upregulated DEGs, while expressing m*HTT* in glia resulted in 3127 downregulated and 3159 upregulated DEGs. There were also DEGs common to both neurons and glia expressing m*HTT*: 1293 downregulated and 1181 upregulated (*Figure 1B*).

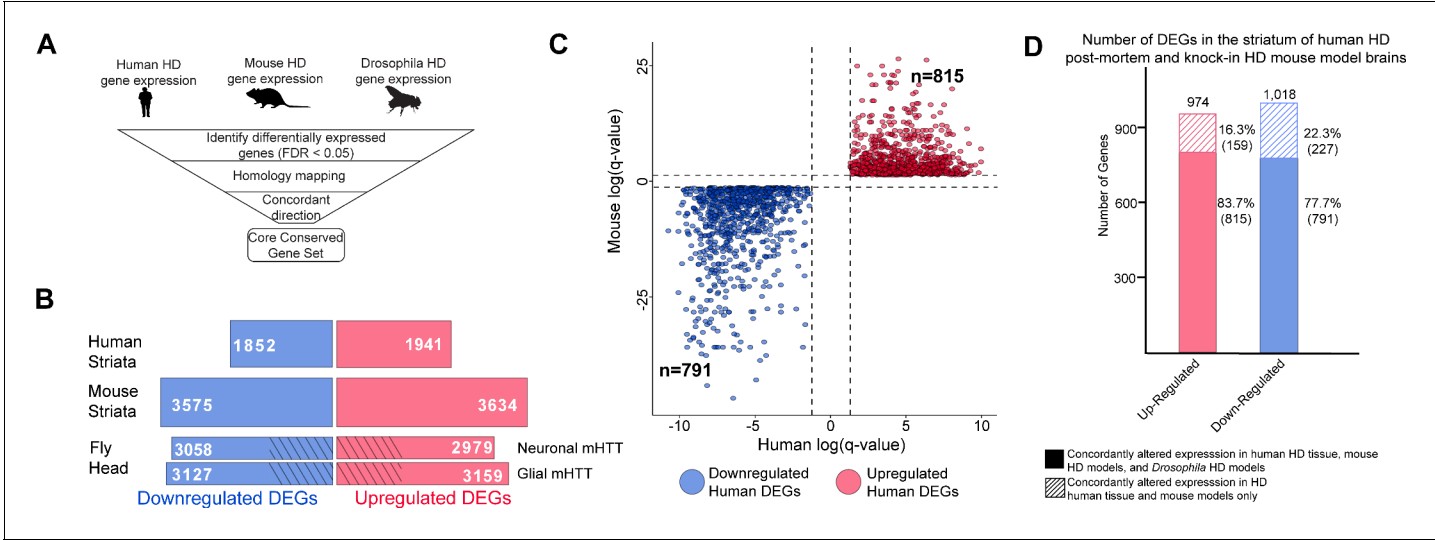

**Figure 1.** Differentially expressed genes (DEGs) in Huntington's disease (HD) human striatal tissue are concordantly altered in mouse and *Drosophila* HD models. (A) Our approach to identifying orthologous genes in tissues from humans, mice, and *Drosophila* with concordant expression changes (i.e., upregulated or downregulated in all three systems) following mutant *Huntingtin* (m*HTT*) expression. (B) The number of DEGs in each species-specific dataset that are downregulated (blue) or upregulated (red) (see Materials and methods). *Drosophila* DEGs were from flies expressing either the N-terminal (*HTT*$^{NT231Q128}$) or full-length m*HTT* (*HTT*$^{FLQ200}$) in neurons (*elav-GAL4*) or glia (*repo-GAL4*). The DEGs in flies are grouped according to the cell type expressing m*HTT* rather than the mHTT model. The cross-hatched regions of the *Drosophila* bars represent DEGs shared between the neuronal and glial sets: 1293 downregulated genes and 1181 upregulated genes. (C) Points in the scatterplot represent human DEGs identified by the strategy outlined in (A) that are concordantly dysregulated across all three species. Red nodes represent upregulated DEGs (n = 815), whereas blue nodes represent downregulated genes (n = 791). The overlap of these concordant DEGs represents approximately 40% of genes with altered expression in the human HD transcriptome that are upregulated (p=6.37×10$^{-158}$) or downregulated (p=1.66×10$^{-165}$). The p-value was calculated using a random background probability distribution over 2 × 10$^5$ random samplings. (D) The stacked bar graph highlights that a large majority of concordant DEGs in human HD striata and knock-in HD mouse models are also concordantly altered in *Drosophila* models of HD.

The online version of this article includes the following source data and figure supplement(s) for figure 1:

**Source data 1.** List of up- and down-regulated differentially expressed genes (DEGs) in humans, mice, and *Drosophila* affected by mutant Huntingtin (mHTT).

**Source data 2.** Lists of up- and down-regulated differentially expressed genes (DEGs) in humans, mice, and *Drosophila* affected by mutant Huntingtin (mHTT) grouped by homology for each *Drosophila* Huntington's disease (HD) model.

**Figure supplement 1.** Expressing mutant Huntingtin (mHTT) in *Drosophila* glia or neurons leads to distinct gene expression profiles.

With these transcriptomic signatures in hand, we were able to compare gene expression profiles across the three species. We focused on genes with significantly altered expression (using a false discovery rate [FDR] < 0.05; see Materials and methods) in the same direction (i.e., upregulated or downregulated) in response to mHTT expression across these three species, including both *Drosophila* HD models. We call genes that meet this criterion concordantly altered DEGs (*Supplementary file 1*).

We compared DEGs using a graph-based approach (see Materials and methods) that allows for evolutionary divergence and convergence, instead of imposing one-to-one relationships. 815 upregulated DEGs observed in HD patient-derived striatal tissue had an orthologous gene in the HD mouse model and at least one *Drosophila* model of HD that was concordantly upregulated. Similarly, 791 DEGs identified in HD patients had an orthologous gene in mouse and *Drosophila* models that was concordantly downregulated (*Figure 1C*). About 40% of the alterations in gene expression in patient striatal samples are concordant with orthologous genes in both *Drosophila* and mice models of HD. To determine whether this result could be an artifact of overlapping a large number of DEGs in each model, we randomly selected and overlapped 815 and 791 orthologous genes across the three species 20,000 times. Based on the resulting distribution, we concluded that the overlap of concordant, orthologous DEGs across the various HD models was not random (p=6.37×10$^{-158}$ and p=1.66×10$^{-165}$, probability distribution test).

To compare the consequence of expressing m*HTT* in glia versus neurons, we recalculated the overlaps between the three species, distinguishing DEGs from the neuron-only and glia-only HTT-expressing *Drosophila*. There were 425 concordantly upregulated and 545 concordantly downregulated DEGs in glia. We also found 522 upregulated DEGs and 453 downregulated specific to neurons. Out of these groups of DEGs, 310 were upregulated and 320 were downregulated in both neurons and glia. To acknowledge the proportion of transcriptional alterations we excluded by specifying concordant expression with the HD *Drosophila* models, we also calculated the overlap between concordant DEGs observed only in striata from HD patients and mice. We found that 83.7% of upregulated DEGs and 77.7% of downregulated DEGs that were altered concordantly in human and mouse HD striata were also concordantly altered in the brains of the neuronal and/or glial HD *Drosophila* models (***Figure 1D***). Of the genes that showed concordantly altered expression only in human and mouse striata, 64 (40%) of the upregulated and 68 (30%) of the downregulated DEGs did not have an ortholog in *Drosophila*.

## Network analysis identifies biological processes disrupted by mHTT toxicity in glia

To investigate the cellular pathophysiology represented by DEGs in neurons and glia, we constructed protein-protein interaction (PPI) networks using the STRING-db database (***Szklarczyk et al., 2015***). The upregulated and downregulated networks of DEGs responding to m*HTT* expression in neurons or glia had a significant PPI enrichment compared to networks constructed from an equivalent number of random genes selected from a whole-proteome background (***Supplementary file 2***). To control for potential artifacts that could arise from using the whole proteome background, we performed a more stringent analysis using only proteins that are found in the striatum (***Al-Ramahi et al., 2018***). Using average node degree and betweenness as proxies for connectivity, we found that the glial and neuronal networks show higher network connectivity than expected by random chance among proteins present in the striatum (***Supplementary file 2***).

This high connectivity suggested that the networks are enriched in specific biological processes and/or pathways. We therefore clustered the glial mHTT response and neuronal mHTT response networks using the InfoMap random walks algorithm (iGraph Package for R and Python) (***Rosvall and Bergstrom, 2007***). Clusters that had fewer than four nodes were filtered out of subsequent analysis. The glial networks formed 23 and 24 clusters for upregulated and downregulated DEGs, respectively. Both the upregulated and downregulated neuronal networks formed 29 clusters. We applied this clustering method to the networks of randomly selected striatal proteins in order to determine the expected number of clusters for networks of a similar size. Both the glial and neuronal networks formed significantly more clusters than would be expected from random selection (***Supplementary file 2***).

To gain insight into biological processes represented by each cluster, we queried the five most significantly enriched terms (FDR < 0.05) using the GO Biological Process and Kyoto Encyclopedia of Genes and Genomes (KEGG) terms within each cluster (***Supplementary file 3***). A synthesis of these terms was used to identify clusters in both the glial and neuronal networks (***Supplementary file 3***, ***Figure 2—figure supplement 1B, C***). We compared the membership within clusters across the glial and neuronal networks using a pairwise hypergeometric test and identified 14 clusters of upregulated DEGs common to both glial and neuronal networks. Similarly, there were 15 clusters of downregulated DEGs common to the both networks (***Figure 2—figure supplement 1A***).

Given the aims of our study, the clusters of DEGs specific to glia (represented by nodes in ***Figure 2***) were of particular interest to us. Six clusters were specifically upregulated in response to m*HTT* expression, enriched in genes involved in transcription and chromatin remodeling, amino acid metabolism, cell proliferation, cytokine signaling/innate immunity, arachidonic acid metabolism, and steroid synthesis (***Figure 2A***). Six clusters were downregulated in response to glial m*HTT* expression, containing genes involved in synapse assembly, calcium ion transport, immune system regulation, phagocytosis, mRNA processing, and fatty acid degradation (***Figure 2B***).

We applied the same network analysis to genes that had concordantly altered expression in HD patient striata and HD mouse model striata but not in HD *Drosophila* models (***Figure 2—figure supplement 2A***). We observed that clusters comprising DEGs specific to the HD patients and the mouse models were functionally related to DEGs in both the glial and neuronal networks (***Figure 2—figure supplement 2B***).

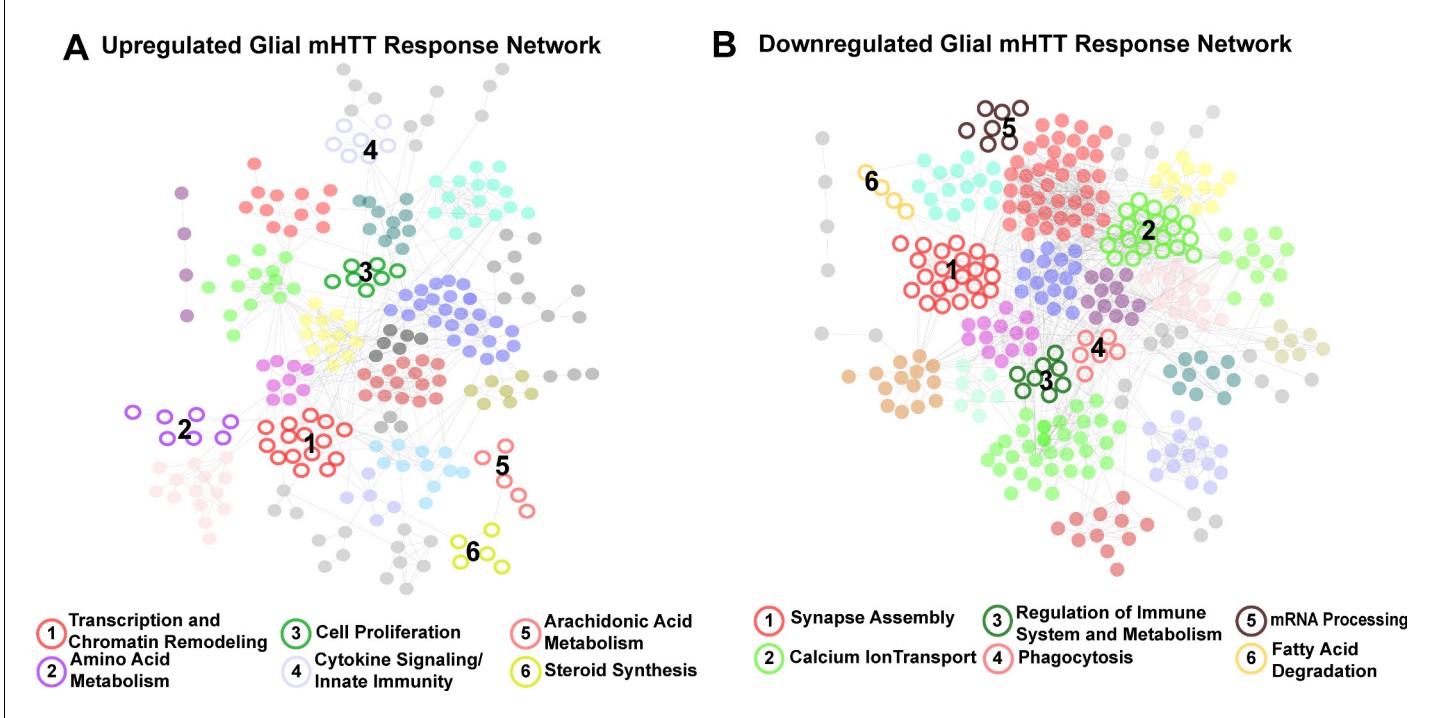

**Figure 2.** Clusters of concordant differentially expressed genes (DEGs) between human and mouse Huntington's disease (HD) striata and *Drosophila* expressing mutant Huntingtin (mHTT) in glia. Clustered protein-protein interaction (PPI) networks of DEGs (STRING-db) that have higher (**A**) or lower (**B**) concordant expression in HD human tissue, an allelic series of knock-in HD mouse models, and *Drosophila* expressing mHTT (HTT$^{NT231Q128}$ or HTT$^{FLQ200}$) in glia. Clusters of DEGs (nodes) that were dysregulated in response to mHTT expression in glia are numbered and represented by open circles. Annotations listed below each network correspond to each numbered cluster and represent a synthesis of the top five most significantly enriched GO Panther Biological processes and Kyoto Encyclopedia of Genes and Genomes (KEGG) terms with a false discovery rate (FDR) < 0.05 (*Supplementary file 2*). Nodes represented by solid circles were dysregulated in response to mHTT expression in glia but are also significantly similar in gene membership to clusters of DEGs in response to mHTT expression in neurons (*Figure 2—figure supplement 1*, hypergeometric test, p<1×10$^{-5}$).

The online version of this article includes the following source data and figure supplement(s) for figure 2:

**Source data 1.** Lists of human Huntington's disease (HD) differentially expressed genes (DEGs) (Entrez IDs) concordantly dysregulated in mouse and *Drosophila* HD models.

**Source data 2.** List of proteins expressed in the human striata.

**Figure supplement 1.** Network of differentially expressed genes (DEGs) responding concordantly to mutant *Huntingtin* (mHTT) expression in glia or neurons.

**Figure supplement 2.** Network of differentially expressed genes (DEGs) concordantly altered in human Huntington's disease (HD) tissue and HD mouse models, but not in *Drosophila* HD models.

## Distinguishing glia-specific gene expression alterations from bulk tissue profiles

Gene expression data from bulk tissue does not provide the resolution required to define cell-autonomous gene expression alterations resulting from mHTT toxicity. Therefore, we compared DEGs (false discovery rate [FDR] < 0.1) in human embryonic stem cells from individuals with HD (carrying 40–48 CAG repeats) with healthy embryonic stem cells that have been differentiated into either CD140+ oligodendrocyte progenitor cells (OPCs) or CD44+ astrocyte progenitor cells (APCs) (*Osipovitch et al., 2019*). We compared the resulting list of DEGs identified in the HD OPCs (1439 genes) and HD APCs (193 genes) to the list of conserved HD DEGs from flies expressing mHTT in glia.

We identified 46 upregulated and 91 downregulated DEGs in common (*Figure 3A*). APCs had 4 upregulated and 12 downregulated genes in common. We next asked whether any clusters in the fly glial networks were enriched in genes dysregulated in HD OPCs or APCs. The Synapse Assembly cluster (*Figure 2B*) was significantly enriched in genes with reduced expression in HD OPCs (Fisher's



**Figure 3.** Reducing the expression of Synapse Assembly cluster genes in glia mitigates mutant Huntingtin (mHTT)-induced behavioral impairments. (**A**) Overlaps between concordant differentially expressed genes (DEGs) from the cross-species analysis defined as responding to mHTT expression in glia and DEGs identified in Huntington's disease (HD) human embryonic stem cells (hESCs) that have been differentiated into either CD140 + oligodendrocyte progenitor cells (OPCs) or CD44+ astrocyte progenitor cells (APCs) (*Osipovitch et al., 2019*). (**B**) Model placing Synapse Assembly cluster proteins into cellular context. The Synapse Assembly cluster was significantly enriched for DEGs in HD OPCs (Fisher's exact test, p<0.001). Only one gene, *NRXN3*, was upregulated in HD OPCs compared to controls (upward red triangle); the rest (*AGAP2, GRM1, LRRTM1, EPB41L2, DLGAP3,* and *SYT13*) were downregulated (downward blue triangles). (**C**) Heatmap representing genes with lower expression in HD OPCs compared to controls (presented as LogFC) that belong to the Synapse Assembly cluster. Each row is one downregulated gene; each column is a different HD human embryonic stem cell line, with CAG repeat length ranging from 40 to 48, compared to respective controls (*Osipovitch et al., 2019*). (**D**) Behavioral assessment of fruit flies that express mHTT only in glia, after reducing the expression of the overlapping DEGs in HD OPCs and the Synapse Assembly cluster. Plots show climbing speed as a function of age. ***p<0.001 between positive control and experimental (by linear mixed effects model and post-hoc pairwise comparison; see Materials and methods). Points and error bars on the plot represent the mean ± SEM of the speed for three technical replicates. Each genotype was tested with 4–6 replicates of 10 animals. Modifying alleles in (**D**) are listed in the Key resources table. Additional climbing data for these genes can be found in *Figure 3—figure supplement 1A,* and a summary of statistical analysis for this data can be found in

*Figure 3 continued*

***Supplementary file 4***. Control climbing data for these alleles can be found in ***Figure 3—figure supplement 1B***. *Drosophila* genotypes: positive control (*w^{1118}*;*UAS- non-targeting hpRNA/+; repo-GAL4,UAS-HTT^{NT231Q128}/+*), treatment control (*w^{1118}; repo-GAL4,UAS- HTT^{NT231Q128}/UAS-siHTT*), and experimental (*w^{1118}; repo-GAL4,UAS- HTT^{NT231Q128}/modifier*).

The online version of this article includes the following source data and figure supplement(s) for figure 3:

**Source data 1.** Raw behavioral data for *Drosophila* expressing mutant *Huntingtin* (m*HTT*) in glia following reduced expression of synaptic genes.

**Figure supplement 1.** Suppressors of glial mutant Huntingtin (mHTT)-induced behavioral impairments among differentially expressed genes (DEGs) in the Synapse Assembly cluster.

**Figure supplement 1—source data 1.** Raw behavioral data for *Drosophila* expressing mutant *Huntingtin* (m*HTT*) in glia following reduced expression of synaptic genes, expanded number of genes, and alleles.

exact test, p<0.001), including *SYT13, LRRTM1, GRM1, EPB41L2, DLGAP3,* and *AGAP2*; the only gene of this cluster that was upregulated in HD OPCs was *NRXN* (***Figure 3B, C***).

In sum, by using a comparative, network-based analysis of the HD transcriptome, we associated dysregulation of several biological processes with the expression of m*HTT* in glia. Layering the gene expression profile of homogenous glial populations affected by mHTT onto these networks, we were able to extract from the bulk-tissue analysis a cluster of genes related to synaptic assembly that are altered in response to glial mHTT toxicity.

## Downregulation of synapse assembly genes is compensatory in HD

The next question we sought to answer is whether changes in expression of synaptic assembly genes are compensatory or pathogenic. We reasoned that if lowering the expression of a downregulated HD DEG aggravated mHTT-induced toxicity, then the downregulation of that gene is pathogenic. Conversely, if reducing the expression of a DEG led to an improvement in HD-related phenotypes, we considered that reduction to be compensatory. We previously used this approach, which takes advantage of the genetic tractability of *Drosophila* and the availability of high-throughput behavioral screening as a proxy for neurological function, to discover modifier genes that reduce HTT protein levels in HD patient cells (***Al-Ramahi et al., 2018***). Here we assessed the effect of various genetic changes in the same group of animals over time, following the expression of mHTT in either glia, neurons, or both cell types. We used a custom, robotic assay system that video-records flies climbing upwards to the top of a vial after being knocked to the bottom (negative geotaxis) to track the behavior of individual *Drosophila* in real time and measure several motor metrics including speed (see Materials and methods). Healthy flies reliably climb to the top at a steady rate until the effects of aging gradually reduce their speed. In contrast, animals expressing mHTT specifically in glia or neurons show much more rapid, if still age-dependent, loss of climbing speed compared to animals expressing a non-targeting hairpin RNA (hpRNA). While we only focus on the effect of these genetic perturbations on speed, we also observe impairments in coordination, balance, and direction (output as number of turns and stumbles) in *Drosophila* expressing mHTT (data not shown).

The expression of *SYT13, LRRTM1, GRM1, EPB41L2, DLGAP3,* and *AGAP2* is reduced in HD OPCs derived from human embryonic stem cells (***Osipovitch et al., 2019***), which is consistent with the expression patterns we observed in patient-derived striatal tissue, knock-in mouse model striatal tissue, and in neuronal tissue from *Drosophila* expressing mHTT in glia (***Figure 3C***). We performed genetic perturbation analysis on the *Drosophila* orthologs of these genes to assess whether their downregulation was pathogenic or compensatory in glia. Diminishing expression of the *Drosophila* orthologs of these six genes mitigated the behavioral deficits induced by mHTT expression in glia (***Figure 3D***, additional controls in ***Figure 3—figure supplement 1B***). We concluded that reduced expression of these genes is a compensatory response to mHTT expression in glia.

There were additional protein interactors in Synapse Assembly whose expression was not altered in the HD-affected OPCs or APCs compared with controls but that were nonetheless downregulated across all three HD models. In our behavioral assay, reducing expression of these interactors, including *NLGN3, NLGN4X, HOMER1,* and *SLITRK5,* was also protective against glial mHTT toxicity (***Figure 3—figure supplement 1A***, ***Supplementary file 4***; additional controls in ***Figure 3—figure supplement 1B***).

In sum, comparative transcriptomic analysis indicated that genes within the Synapse Assembly cluster are associated with the glial response to HD, and the high-throughput behavioral assay further defined this response as compensatory.

## Decreasing neurexin expression in glia mitigates mHTT-induced pathogenesis in both neurons and glia

*NRXN3* was identified as a DEG in both our cross-species comparative transcriptomic analysis and in the gene expression profile of the HD glial progenitor population. *NRXN3* expression was lower in the bulk HD transcriptome across species compared to their respective controls, but it was more highly expressed in the HD OPCs than in controls. This discordance between the bulk and single-cell-type gene expression profiles might be a result of time-dependent changes in gene expression as neurons age, but it prevented us from classifying the *NRXN3* expression changes as being compensatory or pathogenic. We were particularly interested in neurexins, including NRXN3, because they mediate contact between pre- and post-synaptic neurons (*Ushkaryov et al., 1992*; *Zeng et al., 2007*).

We therefore asked whether downregulation of *Drosophila NRXN3* (d*NRXN3*, also known as nrx-1) is damaging or protective when both neurons and glia express m*HTT*. In the *Drosophila* behavioral assay, heterozygous loss of d*NRXN3* function in animals expressing m*HTT* in both neurons and glia mitigated mHTT toxicity and improved behavior (*Figure 4A*, left panel). Reproducing this experiment with flies expressing m*HTT* only in glia yielded the same benefit (*Figure 4A*, middle panel). The obvious next question, given its canonical role in neuron-neuron contact, was whether d*NRXN3* heterozygosity would protect against mHTT pathogenesis in neurons. Interestingly, the answer was no (*Figure 4A*, right panel). Consistent with this, glia-specific knockdown of d*NRXN3* (using the *repo-GAL4* driver) mitigated mHTT toxicity in glia (*Figure 4B*, left panel), but neuron-specific knockdown (using the *elav-GAL4* driver) of d*NRXN3* did not mitigate mHTT toxicity in neurons (*Figure 4B*, right panel). In sum, reducing d*NRXN3* in both neurons and glia protects against glial pathogenesis—and the combination of neuronal and glial pathogenesis—but not neuronal pathogenesis. This implies that mHTT disrupts some aspect of glial-neuronal interaction that is driven by the glia since lowering expression of d*NRXN3* in glia is necessary and sufficient to mitigate behavioral impairments caused by m*HTT*.

To investigate whether *Nrxn3* is expressed in astrocytes in the striatum of HD mice, we performed in situ hybridization (ISH) in coronal sections of striatal tissue taken from a mouse model of HD (*Hdh^{zQ175/+}*) to probe *Nrxn3* mRNA. *Nrxn3* was expressed in striatal astrocytes (*Figure 4C, D*, *Figure 4—figure supplement 1*). In conclusion, modulating the expression genes other than m*HTT* in glia could be an effective strategy for ameliorating HD-induced central nervous system (CNS) dysfunction.

## Reducing SERPINA1 function mitigates behavioral impairments in neurons and glia, and lowers mHTT protein levels

We were curious to identify modifiers that concordantly affect mHTT-induced pathogenesis in both neurons and glia as these might be particularly attractive therapeutic targets for HD. We were particularly interested to discover whether any such shared modifiers exert their effect by reducing mHTT levels, which is considered a promising approach to therapy (*Al-Ramahi et al., 2018*; *Barker et al., 2020*; *Caron et al., 2020*; *Li et al., 2019*; *Tabrizi et al., 2019*; *Wang et al., 2014*; *Wood et al., 2018*; *Yamamoto et al., 2000*; *Yao et al., 2015*). We therefore again integrated network analysis with high-throughput experimentation.

Genes were sampled from both the neuronal and glial mHTT response networks by prioritizing those candidates with high centrality (calculated as a cumulative rank-score of node betweenness and node degree) within each cluster. When available, we used alleles that perturb the expression or activity of the *Drosophila* orthologs in the same direction as the gene expression change in the HD patient population (*Figure 5A*). We screened 411 alleles, representing 248 *Drosophila* genes homologous to 211 human genes, for perturbations that improve the age-dependent behavior of *Drosophila* expressing m*HTT* in neurons or glia (*Supplementary file 5*). Alleles that ameliorated neuronal or glial function were verified in a subsequent trial in animals expressing m*HTT* across the CNS (in both neurons and glia). In all, we identified 25 genes with altered expression in HD that suppressed

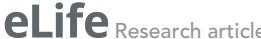

**Figure 4.** Glia-specific d*NRXN3* knockdown mitigates impairments caused by mutant *Huntingtin* (m*HTT*) expression. (**A**) Behavioral assays (climbing speed as a function of age) showing that d*NRXN3* heterozygous loss of function (LOF) ameliorates behavioral impairments caused by expression of m*HTT* in both neurons and glia and in glia alone, but not in neurons alone. (**B**) Glia-specific d*NRXN3* knockdown mitigates behavioral impairments caused by m*HTT* expressed solely in glia; however, neuron-specific knockdown of *dNRXN3* does not affect impairments induced by m*HTT* expressed solely in neurons. \*\*\*$p<0.001$ between positive control and experimental by linear mixed effects model and post-hoc pairwise comparison (see Materials and methods). Points and error bars on the plot represent the mean ± SEM of the speed for three technical replicates. Each genotype was tested with 4–6 replicates of 10 animals. A full summary of the statistical analysis for this data can be found in *Supplementary file 4*. Control climbing data for these alleles can be found in *Figure 3—figure supplement 1B*. (**C**) Astrocytes in the striatum of 6-month-old knock-in HD mice ($Hdh^{zQ175/+}$) expressing *Nrxn3*. In situ probe for *Nrxn3* mRNA is in red (appears magenta when overlapping with the DAPI channel), astrocytes are immunostained using an antibody specific for glial fibrillary acidic protein (GFAP) in green, and DAPI in blue. Image was taken at ×63 magnification using a Leica SP8 confocal microscope. Scale bar (in white on the bottom right) represents 50 µm. 3/5 (60%) of astrocytes in this field appear *Nrxn3* positive. (**D**) Magnified image of the astrocyte highlighted in the white box in (**C**). White arrows indicate yellow puncta where *Nrxn3* mRNA localizes to astrocytes. Scale bar (in white on the bottom right) represents 5 µm. See *Figure 4—figure supplement 1* for additional images and quantification of *Nrxn3* in situ signal in striatal astrocytes in $Hdh^{zQ175/+}$ mice. *Drosophila* genotypes: d*NRXN3* LOF allele ($y^1$ $w*$; Mi{$y^{+mDint2}$=MIC}nrx-1$^{MI02579}$ or nrx-1$^{LOF}$, BDSC: 61696), dNRXN3 RNAi allele (*UAS-nrx-1*$^{hpRNA}$, VDRC: 36326), neuronal and glial Huntington's disease (HD) model with d*NRXN3* mutant (*elav*$^{c155}$-GAL4/ $y^1$ $w*$; repo-GAL4,UAS-HTT$^{NT231Q128}$/Experimental allele), glial HD model with dNRXN3 mutant ($w^{1118}$/$y^1$ $w*$; repo-GAL4,UAS-HTT$^{NT231Q128}$/ Experimental allele), and neuronal model with dNRXN3 mutant (*elav*$^{c155}$-GAL4/$y^1$ $w*$; UAS-HTT$^{NT231Q128}$/Experimental allele).

The online version of this article includes the following source data and figure supplement(s) for figure 4:

**Source data 1.** Raw behavioral data for *Drosophila* expressing mutant *Huntingtin* (m*HTT*) following reduced expression of d*NRXN*.

**Figure supplement 1.** Additional images of astrocytes expressing *Nrxn3* in the striatum of $Hdh^{zQ175/+}$ mice and quantification.

**Figure 5.** Compensatory and pathogenic gene expression changes shared by neurons and glia in response to mutant *Huntingtin* (m*HTT*) expression. (A) Our approach for identifying modifiers of mHTT-induced behavioral impairments common to both neurons and glia. Genes that were central to their respective clusters were prioritized and manipulated in *Drosophila* expressing m*HTT* (*HTT^NT231Q128*) in either neurons (*elav-GAL4*) or glia (*repo-GAL4*). (B) Red bars represent the percent improvement in behavior over a 9-day trial compared to positive control (non-targeting hpRNA) in *Drosophila* expressing *mHTT* in neurons and glia, after we antagonized pathogenic gene expression changes. (C) Green bars represent the percent improvement in behavior over a 9-day trial compared to control (see B), after we mimicked compensatory gene expression alterations. In (B) and (C), the top black bars represent the effect of directly targeting the m*HTT* transgene using a small interfering RNA (siRNA). Arrowheads indicate the direction of the conserved, concordant altered expression for each gene as a result of m*HTT* expression in humans, mice, and *Drosophila*. Behavioral assay graphs corresponding to the data presented in (B) and (C) can be found in *Figure 5—figure supplement 1A*. Corresponding statistical analysis for (B) and (C) can be found in *Supplementary file 6*. Corresponding controls for behavioral data can be found in *Figure 5—figure supplement 1B, C*. *Drosophila* genotypes: positive control (*elav^c155-GAL4/w^1118;UAS- non-targeting hpRNA/+; repo-GAL4,UAS-HTT^NT231Q128/+*), treatment control (*elav^c155-GAL4/w^1118; repo-GAL4, UAS- HTT^NT231Q128/UAS-siHTT*), and experimental (*elav^c155-GAL4/w^1118; repo-GAL4, UAS- HTT^NT231Q128/modifier*). The online version of this article includes the following source data and figure supplement(s) for figure 5:

**Source data 1.** Numerical data for bar charts summarizing the improvement in behavior in *Drosophila* expressing mutant *Huntingtin* (m*HTT*) in neurons and glia by manipulating common pathogenic and compensatory alterations.

**Figure supplement 1.** Genetic modifiers suppress behavioral impairments caused by mutant *Huntingtin* (m*HTT*) expression in neurons and glia.

**Figure supplement 1—source data 1.** Raw behavioral data for *Drosophila* expressing mutant *Huntingtin* (m*HTT*) in neurons and glia by manipulating common pathogenic and compensatory alterations.

mHTT-induced behavioral deficits in neurons, glia, or both (*Figure 5B, C*, *Figure 5—figure supplement 1*, *Supplementary file 6*).

Many of the modifiers common to neuronal and glial mHTT-induced dysfunction are involved in the regulation of the actin cytoskeleton (*RHOC, TIAM1, ENAH,* and *CFL2*), vesicular trafficking (*SNAP23, SNX9,* and *SNX18*), and inflammation (*JUN, GTF3A,* and *ATF3).* Multiple reports have implicated components of these pathways in the pathogenesis of not only HD, but in other neurodegenerative disorders as well (*Al-Ramahi et al., 2018*; *Bardai et al., 2018*; *Bondar et al., 2018*). We previously established an axis of genes with altered expression that regulate actin cytoskeleton and inflammation pathways driving forward HD pathogenesis (*Al-Ramahi et al., 2018*). Our current results would indicate that these pathways are not only critical to disease progression in neurons, but also in glia.

We previously observed that reducing the activity of RAC GTPase, a regulator of the actin cytoskeleton, and inflammation mediating nuclear factor *kappa*-light-chain-enhancer of activated *B* cells (*NF Kappa-B*) ameliorated pathogenesis by lowering mHTT protein levels through the activation of autophagy (*Al-Ramahi et al., 2018*). Thus, in a secondary screen we tested whether these disease modifiers common to both neurons and glia exerted their beneficial effects by lowering levels of the mutant HTT protein.

We collected protein lysates from *Drosophila* expressing m*HTT* across the CNS that also bore alleles that suppressed mHTT-induced behavioral deficits in both neurons and glia. We assessed the quantity of mHTT protein in these lysates by western blot, comparing experimental (candidate modifiers) and control animals (carrying a non-targeting hpRNA). This secondary screen identified *Spn42De* as a modifier whose knockdown lowered mHTT levels. *Spn42De* is one of the four *Drosophila* homologues of human *SERPINA1* (which encodes alpha-1-antitrypsin, a member of a large group of protease inhibitors). *Spn42De*, human *SERPINA1,* and mouse *Serpina1* are all upregulated in HD, and they are part of the Wound Healing and Inflammation cluster in both the neuronal and glial mHTT response networks (*Figure 2—figure supplement 1C*). Knockdown of *Spn42De* (henceforth *dSERPINA1*) in *Drosophila* expressing m*HTT* in both neurons and glia mitigated behavioral impairments (*Figure 6A*). In independent immunoblots, *dSERPINA1* knockdown consistently reduced mHTT protein levels in lysates extracted from the heads of *Drosophila* expressing m*HTT* in both neurons and glia (*Figure 6B, C*). As a control, we performed immunoblot analysis of lysates from a green fluorescent protein (GFP) reporter line to ensure that this allele of *dSERPINA1* did not reduce the function of the *GAL4-UAS* system (*Figure 6—figure supplement 1*).

To validate this observation across model systems, we performed homogenous time-resolved fluorescence (HTRF) on $Hdh^{Q111/Q7}$ mouse striatal cell lysates that were treated with either a pool of non-targeting scramble small interfering RNAs (siRNAs), a pool of siRNAs against *Htt*, or a pool of siRNAs against *Serpina1a* (the murine ortholog of *SERPINA1*). *Serpina1a* knockdown significantly reduced mHTT signal (*Figure 6D*). Knockdown of *SERPINA1* thus protected against mHTT toxicity in neurons and glia by reducing levels of mutant HTT. Verifying this effect in multiple model organisms increases confidence in this observation and suggests that *SERPINA1* could potentially prove useful as a target for treating HD. Interestingly, *SERPINA1* expression is low in the healthy brain but it is upregulated in several disease conditions, consistent with a potential role in neuroinflammation (*Abu-Rumeileh et al., 2020*; *Cabezas-Llobet et al., 2018*; *Gollin et al., 1992*; *Peng et al., 2015*). We found increased Serpina1a protein staining in the striatum of $Hdh^{zQ175/+}$ compared to wildtype mice at 8.5 months (*Figure 6—figure supplement 2*), confirming its upregulation from the transcriptomic data. Previously we had shown that other genes in the subnetwork implicated in neuroinflammation can be manipulated to lower mHTT protein levels (*Al-Ramahi et al., 2018*). *SERPINA1* may thus warrant investigation as a target for other neurological disorders as well.

## Discussion

We found a high degree of overlap of DEGs across tissues from human HD brains, brains of HD mice, and flies that express m*HTT* in glia. This may seem unexpected given obvious differences between vertebrate and *Drosophila* glia, such as a lack of documented microglia or distinct morphology of endothelial/glial cells forming the blood-brain barrier in *Drosophila* (*Freeman and Doherty, 2006*). Our observations are however consistent with previous evidence that *Drosophila* glia perform many of the same functions as mammalian astrocytes, oligodendrocytes, endothelial cells, and

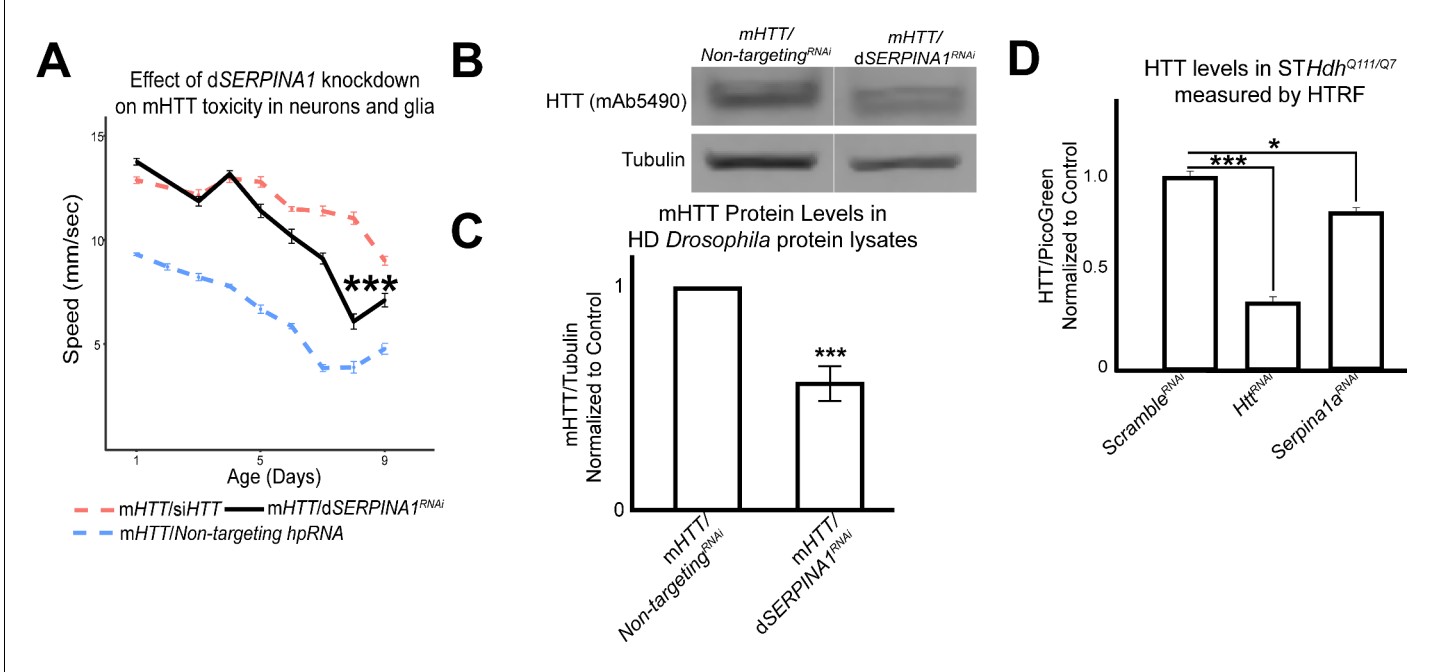

**Figure 6.** Antagonizing the pathogenic overexpression of *SERPINA1* in neurons and glia mitigates mutant *Huntingtin* (mHTT)-induced behavioral impairments and lowers mHTT protein levels in *Drosophila* and Huntington's disease (HD) mouse striatal cells. (**A**) Behavioral assays following knockdown of *dSERPINA1* in *Drosophila* expressing m*HTT* in neurons and glia. *** indicates p<0.001 by linear mixed effects model and post-hoc pairwise comparison between positive control and experimental animals. Points and error bars on the plot represent the mean ± SEM of three technical replicates. Each genotype was tested with 4–6 replicates of 10 animals. (**B**) Representative western blot showing lower levels of mHTT following knockdown of *dSERPINA1* in *Drosophila* expressing m*HTT* in neurons and glia. (**C**) Quantification of five independent immunoblots showing the effect of *dSERPINA1* knockdown on mHTT levels in *Drosophila* head protein lysates. ***p<0.001 between positive control and d*SERPINA1* knockdown by one-way t-test. (**D**) Quantification of HTT protein levels in HD mouse striatal-derived cells (ST*Hdh*$^{Q111/Q7}$) measured by homogenous time-resolved fluorescence (HTRF) following treatment with a pool of scramble small interfering RNAs (siRNAs) (negative control), a pool of siRNAs against *Htt*, and a pool of siRNAs against *Serpina1a*. Quantification is presented as a ratio of the emission signal from the fluorescent D2 dye (HTT)/PicoGreen (number of cells per well). n = 9 for each treatment group. *p<0.05 and ***p<0.001 between genotypes by Fisher's Least Significant Difference (LSD) test. *Drosophila* genotypes: *dSERPINA1* RNAi allele (*UAS-Spn42De*$^{hpRNA}$, VDRC: 102622), positive control (*elav*$^{c155}$-*GAL4/w*$^{1118}$;*UAS- non-targeting hpRNA/+; repo-GAL4,UAS-HTT*$^{NT231Q128}$/+), treatment control (*elav*$^{c155}$-*GAL4/w*$^{1118}$; *repo-GAL4, UAS- HTT*$^{NT231Q128}$/*UAS-siHTT*), and dSERPINA1 experimental (*elav*$^{c155}$-*GAL4/w*$^{1118}$;*UAS-Spn42De*$^{hpRNA}$/+; repo-GAL4, UAS-HTT$^{NT231Q128}$/+).

The online version of this article includes the following source data and figure supplement(s) for figure 6:

**Source data 1.** Raw behavioral data for *Drosophila* expressing mutant *Huntingtin* (m*HTT*) in neurons and glia following knockdown of Spn42De.

**Source data 2.** Summary of numerical and raw data for western blots of protein lysates from *Drosophila* heads expressing mutant *Huntingtin* (m*HTT*) in neurons and glia with knockdown of Spn42De.

**Source data 3.** Summary of numerical and raw data for homogenous time-resolved fluorescence (HTRF) for protein lysates of ST*Hdh*$^{Q111/Q7}$ treated with pooled small interfering RNAs (siRNAs) against Serpina1a.

**Figure supplement 1.** d*SERPINA1* knockdown does not reduce the expression of the *GAL4-UAS* system.

**Figure supplement 1—source data 1.** Summary of numerical and raw data for western blots of protein lysates from *Drosophila* heads expressing *mCD8::GFP* in neurons with knockdown of Spn42De.

**Figure supplement 2.** Serpina1a protein accumulates in the striata of *Hdh*$^{zQ175/+}$ mice compared to wildtype littermates.

**Figure supplement 2—source data 1.** Summary of numerical and raw data for immunohistochemistry analysis of Serpina1 staining in 8.5-month-old mouse striata from wildtype and zQ175 mice.

microglia including phagocytosis (*Chung et al., 2020*; *Freeman, 2015*; *Freeman and Doherty, 2006*; *Ziegenfuss et al., 2012*). In fact, the overlap of concordant DEGs between mammalian and *Drosophila* glia may be underestimated in our analysis because it was limited to CD44+ and CD140 + cells from human embryonic stem cell-derived glial progenitors and therefore we may have missed DEG overlaps from other glial types, or from more mature state of oligodendrocytes or astrocytes.

Several studies have also shown that wildtype glial cells ameliorate disease when transplanted into HD mice, and mHTT exerts a deleterious effect on glial development and function, which in turn

influences HD pathogenesis (*Benraiss et al., 2016*; *Bradford et al., 2009*; *Garcia et al., 2019*; *Huang et al., 2015*; *Osipovitch et al., 2019*). More recently, it was discovered that transcription factors involved in glial differentiation and myelin synthesis are downregulated in glial progenitor cells (*Osipovitch et al., 2019*). Yet despite this progress, the overall contributions of glial genes to synaptic impairments and other key neurodegenerative pathologies remain poorly understood. The genetic malleability of *Drosophila* enabled us to thoroughly examine the neuron-glia interface from both the glial and the neuronal directions.

Synaptic dysfunction is a common theme among many neurodegenerative disorders (*McInnes et al., 2018*; *Phan et al., 2017*; *Prots et al., 2018*). While it is clear that the dysfunction of the glia-synapse interface is central to the pathophysiology of neurodegeneration (*Filipello et al., 2018*; *Garcia et al., 2019*; *Lian et al., 2015*; *Litvinchuk et al., 2018*), the underlying mechanisms remain underexplored relative to the interactions between pre- and post-synaptic neurons. Our results support the observation that the expression of m*HTT* in glia is sufficient to drive synaptic dysfunction (*Wood et al., 2018*). In HD, pre-synaptic neurons release elevated levels of glutamate into the synapse, driving medium spiny neurons (MSNs) into excitotoxicity (*Estrada Sánchez et al., 2008*; *Hong et al., 2016*). Hyperactivity of receptors at the post-synaptic densities sensitizes MSNs to excitotoxicity, further contributing to neurodegeneration (*Estrada Sánchez et al., 2008*). Astrocytic m*HTT* expression may contribute to neuronal excitotoxicity by elevating levels of glutamate, potassium, and calcium at the synapse (*Garcia et al., 2019*; *Jiang et al., 2016*; *Tong et al., 2014*).

Modifiers of mHTT-induced pathogenesis identified in our study, such as metabotropic glutamate receptors and the scaffold protein HOMER1, regulate calcium and glutamate signaling in astrocytes (*Buscemi et al., 2017*; *Spampinato et al., 2018*). Reducing the expression of these genes could prevent excess calcium and glutamate from accumulating at the synapse. Indeed, we previously found that HD neurons downregulate the expression of genes involved in calcium signaling in an effort to compensate for HD pathogenesis (*Al-Ramahi et al., 2018*). Glial calcium signaling can also influence neuronal activity, however, at the neuronal soma (*Weiss et al., 2019*). In *Drosophila*, cortical glia modulate neuronal activity through potassium buffering, a process that is regulated by calcium-mediated endocytosis of potassium channels (*Weiss et al., 2019*). Glia can also physically disrupt synapses in disease states: Förster resonance energy transmission in vivo revealed that, in HD, the distances between astrocytes and pre-synaptic neurons are increased at the cortico-striatal circuit (*Octeau et al., 2018*). Thus, knocking down the genes in the Synapse Assembly cluster could reduce physical interaction between glia and synapses, promoting normal synaptic function.

If in HD synapses grow more fragile and fewer in number as the disease progresses, why would downregulating the expression of glial genes required for synapse formation and function be protective? We postulate it is for the same reason that downregulating calcium-signaling genes is compensatory (*Al-Ramahi et al., 2018*): the brain is attempting to protect against the excitotoxicity described above. Mutant HTT disrupts neuronal development (*Ring et al., 2015*) and skews embryonic neurogenesis toward producing more neurons (*Barnat et al., 2020*); by the time HD mutation carriers reach the age of 6 years, they have greatly enlarged striata and functional hyperconnectivity to the cerebellum (*Tereshchenko et al., 2020*). The more hyperconnected, the more abrupt the loss of these connections, and the more rapid the striatal atrophy that follows *Tereshchenko et al., 2020*. The hyperfunction of a given brain region puts considerable strain on the circuit, and it seems that over the course of a lifetime, the brain keeps trying to compensate for the abnormalities that arise at different stages of HD. The recent observation that deletion of astrocytic neurexin-1α attenuates synaptic transmission but not synapse number supports this hypothesis (*Trotter et al., 2020*).

We do not think that the protection provided by modifiers in this cluster is limited to modulating neurotransmission. In astrocytes, calcium signaling also controls the activity of reactive astrocytes (*Buscemi et al., 2017*). Astrogliosis, or the proliferation of immune active astrocytes, is typically observed at later stages of HD (*Al-Dalahmah et al., 2020*; *Buscemi et al., 2017*). These immune-activated glia not only eliminate synapses (*Liddelow et al., 2017*; *Sofroniew, 2009*) but can also transmit mHTT aggregates through the synapse (*Donnelly et al., 2020*). In *Drosophila*, knockdown of *draper* prevents astrocytic phagocytosis and stops the spread of mHTT protein aggregates from pre-synaptic neurons to the post-synaptic compartment (*Donnelly et al., 2020*; *Pearce et al., 2015*). mHTT protein can also enter the synaptic space by endosomal/lysosomal secretion mediated by Syt7 (*Trajkovic et al., 2017*). In this study, we observed that knockdown of synaptotagmins in *Drosophila* ameliorates glial mHTT-induced dysfunction. Thus, knocking down genes in the Synapse

Assembly cluster could also benefit the circuit by reducing the transmission of aggregated mHTT protein from pre- to post-synaptic neurons.

Intriguingly, loss-of-function variants in *NRXN1-3, NLGN1, NLGN3, DLGAP3,* and *LRRTM1* have been associated with various disorders of synaptic dysfunction, including autism spectrum disorder (ASD), schizophrenia, and obsessive compulsive disorder (OCD) (*Nakanishi et al., 2017*; *Jamain et al., 2003*; *Südhof, 2008*; *Vaags et al., 2012*; *Wang et al., 2018*; *Windrem et al., 2017*). We speculate that the consequences of loss of function of these genes depend on both dosage and context: modest reductions of gene expression can be protective in the context of HD pathogenesis, whereas a more severe loss of function results in ASD and OCD. It is interesting that many HD patients develop schizophrenia-like psychosis, suggesting that the compensatory mechanism at place in HD may eventually lead to schizophrenia-like symptoms (*Connors et al., 2020*; *Tsuang et al., 2018*). Future studies should investigate whether these loss-of-function variants associated with neurodevelopmental and psychiatric disorders alter the age of disease onset in patients with HD. It could be of particular interest to assess if these neurodevelopmental and psychiatric-associated variants ameliorate neurodevelopmental changes observed early in HD or blunt synaptic hyperactivity later in disease.

# Materials and methods

**Key resources table**

| Reagent type (species) or resource | Designation | Source or reference | Identifiers | Additional information |
| --- | --- | --- | --- | --- |
| Antibody | Anti-HTT (mouse monoclonal) | EMD Millipore | mAb5490, RRID:AB_2233522 | WB (1:500) |
| Antibody | Anti-GFP (rabbit polyclonal) | ThermoFisher | A-11122, RRID:AB_221569 | WB (1:1000) |
| Antibody | Anti-HTT (mouse monoclonal) | Novartis | 2B7 | HTRF (0.023 µg/mL) |
| Antibody | Anti-laminC (mouse monoclonal) | Hybridoma Bank | LC28.26, RRID:AB_528339 | WB (1:1000) |
| Antibody | Anti-GFAP (rabbit polyclonal) | DAKO | Z0334, RRID:AB_10013382 | IF (1:500) |
| Antibody | Alpha-tubulin (rabbit polyclonal) | Abcam | EP1332Y, RRID:AB_922700 | WB (1:1000) |
| Antibody | Anti-HTT (mouse monoclonal) | SigmaAldrich | mAb2166, RRID:AB_11213141 | HTRF (1.4 µg/mL) |
| Antibody | Anti-rabbit IgG Alexa 488 (goat polyclonal) | Invitrogen | A-11008, RRID:AB_143165 | IF (1:500) |
| Antibody | Anti-Serpina1a (rabbit polyclonal) | Invitrogen | PA5-16661, RRID:AB_10985745 | IF (1:250) |
| Antibody | RDye 680RD anti-Rabbit IgG (goat polyclonal) | LI-COR Biosciences | 925-68071, RRID:AB_2721181 | WB (1:5000) |
| Antibody | IRDye 800CW anti-Mouse IgG (goat polyclonal) | LI-COR Biosciences | 925-32210, RRID:AB_2687825 | WB (1:5000) |
| Chemical compound, drug | Lipofectamine 2000 | Life Technologies | 11668 | |
| Chemical compound, drug | EDTA-free protease inhibitor | Calbiochem | 539134 | |
| Commercial assay, kit | miRNeasy Mini Kit | Qiagen | 217004 | |
| Commercial assay, kit | Illumina TruSeq Stranded mRNA | Illumina | 20020595 | |
| Commercial assay, kit | Tyramide-Cy3 Plus kit | Perkin Elmer | NEL744001KT | |

*Continued on next page*

Continued

| Reagent type (species) or resource | Designation | Source or reference | Identifiers | Additional information |
|---|---|---|---|---|
| Commercial assay, kit | PicoGreen | Quant-iT PicoGreen dsDNA Assay Kit | P7589 | |
| Cell line (*Mus musculus*) | STHdh*$^{Q111/Q7}$* Cells | Coriell Cell Repositories | CH00096 | |
| Strain, strain background (*Drosophila*) | White mutant, background genotype | Bloomington *Drosophila* Stock Center | 3605 | *w$^{1118}$* |
| Genetic reagent (*Drosophila*) | Non-targeting hpRNA | Vienna *Drosophila* Resource Center | 13974 | |
| Strain, strain background (*Drosophila*) | repo-Gal4 | Bloomington *Drosophila* Stock Center | 7415 | *w$^{1118}$; P{w$^{+m*}$=GAL4} repo/TM3, Sb$^1$* |
| Strain, strain background (*Drosophila*) | elav-Gal4 | Bloomington *Drosophila* Stock Center | 458 | *P{GawB}elav$^{C155}$* |
| Genetic reagent (*Drosophila*) | N-terminal HD model | Botas Laboratory | **Branco et al., 2008** | *UAS-HTT$^{NT231Q128}$/TM6B, tubulin-GAL80* (N-terminal) |
| Genetic reagent (*Drosophila*) | Full-length HD model | Botas Laboratory | This paper | *UAS-HTT$^{FL200Q}$/CyO* (full-length) |
| Genetic reagent (*Drosophila*) | siRNA against human mutant HTT | Botas Laboratory **Kaltenbach et al., 2007** | | *UAS-siHTT* |
| Genetic reagent (*Drosophila*) | Classic *CenG1A* loss-of-function allele | Bloomington *Drosophila* Stock Center | 44301 | *CenG1A$^{LOF}$* or *y$^1$w\*; Mi{MIC}CenG1A$^{MI06024}$* (**Figure 3D**) |
| Genetic reagent (*Drosophila*) | Classic *vlc* loss-of-function allele | Bloomington *Drosophila* Stock Center | 10366 | *vlc$^{LOF}$* or *y$^1$w$^{67c23}$;P{w$^{+mc}$ = lacW}vlc$^{k01109}$/CyO* (**Figure 3D**) |
| Genetic reagent (*Drosophila*) | Classic *trn* loss-of-function allele | Bloomington *Drosophila* Stock Center | 4550 | *trn$^{LOF}$* or *y$^1$w$^{67c23}$;P{w$^{+mc}$ = lacW}trn$^{S064117}$/TM3, Sb$^1$ Ser$^1$* (**Figure 3D**) |
| Genetic reagent (*Drosophila*) | Classic *cora* loss-of-function allele | Bloomington *Drosophila* Stock Center | 9099 | *cora$^{LOF}$* or *P{ry$^{t7.2}$=neoFRT} 43D cora$^{14}$/CyO* (**Figure 3D**) |
| Genetic reagent (*Drosophila*) | RNAi against *Sytbeta* | Vienna *Drosophila* Resource Center | 106559 | *UAS-Sytbeta$^{hpRNA}$* (**Figure 3D**) |
| Genetic reagent (*Drosophila*) | RNAi against *mGluR* | National Institute of Genetics, Japan | 11144 R-3 | *UAS-mGluR$^{RNAi}$* (**Figure 3D**) |
| Genetic reagent (*Drosophila*) | Neuronal mCD8::GFP reporter line | Bloomington *Drosophila* Stock Center | 5146 | *P{w$^{+mW.hs}$=GawB}elav$^{C155}$, P{w$^{+mC}$ = UAS-mCD8::GFP.L} Ptp4E[LL4], P{ry[+t7.2]=hsFLP}1, w\** |
| Genetic reagent (*Drosophila*) | Classical loss of function and overexpression alleles in *Drosophila* | Bloomington *Drosophila* Stock Center | Provided in **Supplementary files 4** and **5** | |
| Genetic reagent (*Drosophila*) | RNAi alleles in *Drosophila* | Vienna *Drosophila* Resource Center | Provided in **Supplementary files 4** and **5** | |
| Genetic reagent (*Drosophila*) | Cytological duplication alleles in *Drosophila* | GenetiVision | Provided in **Supplementary files 4** and **5** | |
| Genetic reagent (*M. musculus*) | Hdh*$^{zQ175}$* Mice | Jackson Laboratories | 027410 | B6J.129S1-Htttm 1Mfc/190ChdiJ |

*Continued on next page*

*Continued*

| Reagent type (species) or resource | Designation | Source or reference | Identifiers | Additional information |
|---|---|---|---|---|
| Recombinant DNA reagent | pMF3 Vector | *Drosophila* Genome Resource Center | 1237 | |
| Software, algorithm | Adept Desktop | Omron | N/A | |
| Software, algorithm | Video Savant | IO Industries | N/A | |
| Software, algorithm | MatLab with Image Processing Toolkit and Statistics Toolkit | MathWorks | https://www.mathworks.com/products/matlab.html | |
| Software, algorithm | RSLogix | Rockewell Automation | N/A | |
| Software, algorithm | Ultraware | Rockewell Automation | N/A | |
| Software, algorithm | Assay Control | SRI International | N/A | |
| Software, algorithm | FastPhenoTrack Vision Processing | SRI International | N/A | |
| Software, algorithm | TrackingServer Data Management | SRI International | N/A | |
| Software, algorithm | ScoringServer Behavioral Scoring | SRI International | N/A | |
| Software, algorithm | Trackviewer Visual Tracking and Viewing | SRI International | N/A | |
| Software, algorithm | Illustrator CC | Adobe | https://www.adobe.com | |
| Software, algorithm | R | R Project for Statistical Computing | https://www.r-project.org/ | |
| Software, algorithm | Fiji | The Fiji Team | https://fiji.sc/ | |
| Software, algorithm | Image Studio Lite | LI-COR Biosciences | https://www.licor.com/bio/image-studio-lite/ | |
| Software, algorithm | Bowtie | *Langmead and Salzberg, 2012* | http://bowtie-bio.sourceforge.net/index.shtml | |
| Software, algorithm | RSEM | *Li and Dewey, 2011* | https://github.com/deweylab/RSEM | |
| Software, algorithm | DESeq2 | *Love et al., 2014* | https://bioconductor.org/packages/release/bioc/html/DESeq2.html | |
| Software, algorithm | DIOPT | *Hu et al., 2011* | https://www.flyrnai.org/cgi-bin/DRSC_orthologs.pl | |
| Software, algorithm | MGI | The Mouse Genome Database | http://www.informatics.jax.org/genes.shtml | |
| Software, algorithm | STRING | *Szklarczyk et al., 2015* | https://string-db.org/ | |
| Software, algorithm | InfoMap | *Rosvall and Bergstrom, 2008* | https://cran.r-project.org/web/packages/igraph/index.html | |
| Software, algorithm | Cytoscape | The Cytoscape Consortium | https://cytoscape.org | |

*Continued on next page*

*Continued*

| Reagent type (species) or resource | Designation | Source or reference | Identifiers | Additional information |
|---|---|---|---|---|
| Transfected construct (*M. musculus*) | AllStars Negative Control siRNA (Scramble) | Qiagen | 1027280 | |
| Transfected construct (*M. musculus*) | *Htt* SMARTPool siRNAs | Horizon Discovery Limited | L-040632-01-0005 | 5′- GAAAUUAAGGUUCUGUUGA-3′<br>5′- CCACUCACGCCAACUAUAA-3′<br>5′- GAUGAAGGCUUUCGAGUCG-3′<br>5′- UAACAUGGCUCAUUGUGAA-3′ |
| Transfected construct (*M. musculus*) | *Serpina1a* SMART Pool siRNAs | Horizon Discovery Limited | L-043380-01-0005 | 5′- GAAUAUAACUUGAAGACAC-3′<br>5′-GGGCUGACCUCUCCGGAAU-3′<br>5′- UGGUAGAUCCCACACAUAA-3′<br>5′- GAAAGAUAGCUGAGGCGGU-3′ |
| Sequence-based reagent | Primers for cloning human *HTT* | This paper | See experimental model detail | Forward<br>5′-gaattcGCACCGACC AAAGAAAGAAC-3′<br>Reverse<br>5′-tctagaGGCAGAAGG TTCACCAGGTA-3′ |
| Sequence-based reagent | Primers for generating in situ probes for mouse *Nrxn3* including RNA polymerase promoter sequences for T3 (forward) and T7 (reverse) | Allen Brain Atlas | https://portal. brain-map.org/ | Forward: 5′-GCGAATTAACCCTCACTAAA GGGTCCTTCCCCTTTCCTCCTAA-3′<br>Reverse: 5′-GCGTAATACGACTCACTATAGG GCAGGCATGCTCTGTACTCCA-3′ |

## Lead contact and material availability

Further information and requests for resources and reagents should be directed to and will be fulfilled by the lead contact, Juan Botas (jbotas@bcm.edu).

## *Drosophila* models

We began with *Drosophila* models expressing either N-terminal human HTT (*HTT^NT231Q128*) or full-length HTT (*HTT^FLQ200*) (*Kaltenbach et al., 2007*; *Romero et al., 2008*). The mHTT was expressed using either a pan-neuronal (*elav*) or a pan-glial driver (*repo*). Mutant strains for screening were obtained from Bloomington *Drosophila* Stock Center, GenetiVision, and the Vienna *Drosophila* Resource Center. All strains were maintained at 18°C in standard molasses, yeast extract, and agar media until their experimental use. For RNA-sequencing, the full-length models were raised at 29°C and the N-terminal models were raised at 28°C. All behavioral experiments were performed on females raised at 28°C.

In *Figure 3D*, we used the following mutants to assess the effect of reduced expression of synaptic genes in mHTT animals on behavior: *UAS-non-targeting^hpRNA* (Vienna *Drosophila* Resource Center, ID:13974), *CenG1A^LOF* or *y^1w*;Mi{MIC}CenG1A^MI06024* (Bloomington *Drosophila* Stock Center, ID: 44301), *vlc^LOF* or *y^1w^67c23;P{w^+mc = lacW}vlc^k01109/CyO* (Bloomington *Drosophila* Stock Center, ID: 10366), *trn^LOF* or *y^1w^67c23;P{w^+mc = lacW}trn^S064117/TM3, Sb^1 Ser^1* (Bloomington *Drosophila* Stock Center, ID: 4550), *cora^LOF* or *P{ry^t7.2=neoFRT}43D cora^14/CyO* (Bloomington *Drosophila* Stock Center, ID: 9099), *UAS-Sytbeta^hpRN A* (Vienna *Drosophila* Resource Center, ID:106559), and *UAS-mGluR^RNAi* (National Institute of Genetics, Japan, ID: 11144-R3).

To generate *Drosophila* that expressed siRNA that knocked down human *HTT* (*UAS-siHTT*), we cloned a 378 bp inverted EcoRI, XbaI fragment of N-terminal Htt into the pMF3 vector (*Drosophila* Genome Resource Center). This fragment maps to base pairs 406–783 of the human mRNA *Huntingtin,* which we cloned using the following primers:

Forward 5′-gaattcGCACCGACCAAAGAAAGAAC-3′

Reverse 5′-tctagaGGCAGAAGGTTCACCAGGTA-3′

We first digested the PCR product with EcoRI and ligated it with itself to obtain inverted repeats. We then digested the inverted repeat with XbaI and pasted the fragment into the pMF3 vector (also cut with XbaI); the resulting plasmid was injected into *Drosophila* embryos using standard methods (*Dietzl et al., 2007*). We validated that this line lowers mHTT levels.

## STHdh<sup>Q111/Q7</sup> mouse striatal cells

Immortalized mouse striatal cells heterozygous for m*HTT* (*STHdh*<sup>Q111/Q7</sup>) were obtained from Coriell Cell Repositories (Camden, NJ) and cultured in DMEM (Life Technologies, cat. no. 11965) supplemented with 10% fetal bovine serum (Life Technologies, cat. no. 10082–147). The cells were tested every two months by a TransDetect PCR Mycoplasma Detection Kit (Transgen Biotech, cat. no. FM311-01) to ensure that they are mycoplasma free. The identity has not been authenticated by STR profiling, but has been validated by western blot, morphology, and phenotypic experiments.

## DEG identification in *Drosophila* HD models

We performed RNA-seq on head tissue collected from *Drosophila* expressing N-terminal (*UAS-HTT*<sup>NT231Q128</sup>) or full-length (*UAS-HTT*<sup>FLQ200</sup>) human mHTT in neurons (*elav-GAL4*) or glia (*repo-GAL4*). For each combination of HD model and driver, RNA-seq was performed at three timepoints to capture the early, middle, and late phases of disease pathogenesis, corresponding to behavioral deficits caused by mHTT-induced neuronal or glial dysfunction. At each timepoint, samples for HD and age-matched controls were collected in triplicate. *Drosophila* expressing the N-terminal construct and corresponding controls were obtained at 7, 9, and 11 days post-eclosion for the neuronal driver, and at 5, 7, and 8 days post-eclosion for the glial driver. *Drosophila* expressing the full-length construct, samples were obtained at 18, 20, and 22 days post-eclosion for both the neuronal and glial driver. For RNA-seq, the neuronal N-terminal, glial N-terminal, and glial full-length model *Drosophila* were raised at 28°C. The neuronal full-length model *Drosophila* were raised at 29°C. For each genotype at each timepoint, we collected an equivalent number of control animals (*elav-GAL4* or *repo-GAL4*) that were raised in the same conditions.

Three replicates of 50 virgin females were collected for each genotype and timepoint. Animals were aged in the appropriate temperature and were transferred to fresh food daily until tissue was harvested. At the selected ages, animals were transferred to 1.5 mL tubes, flash frozen in liquid nitrogen, vigorously shaken, and then sieved to collect 50 heads/genotype/replica (~5 mg tissue/replica). Total RNA was extracted using the miRNeasy Mini Kit (Qiagen cat. no. 210074).

RNA-seq profiling and preprocessing was performed by Q2 Solutions (Morrisville, NC). Samples were converted into cDNA libraries using the Illumina TruSeq Stranded mRNA sample preparation kit (Illumina cat. no. 20020595) and were sequenced using HISeq-Sequencing-2 × 50 bp-PE. Initial analysis was performed using Q2 Solution in-house mRNAv7 pipeline with a median of 49 million actual reads. After adapter sequences were removed, the reads were aligned to the *Drosophila melanogaster* transcriptome using Bowtie version 0.12.9 (*Langmead and Salzberg, 2012*). Expression was quantified using RSEM version 1.1.19, resulting in a median of 11,214 genes and 18,604 isoforms detected (*Li and Dewey, 2011*).

## Homology mapping of HD DEGs by network-based intersection

Three homology maps were constructed to define conserved genes that were concordantly dysregulated in response to mHTT toxicity: a *Drosophila*-human map, a *Drosophila*-mouse map, and a mouse-human map. The *Drosophila*-human map and *Drosophila*-mouse map were both obtained from DIOPT version 6.0.2 (*Hu et al., 2011*). To capture homology that results from evolutionary convergence and divergence, we included lower DIOPT scores between *Drosophila* and mammals instead of fitting one-to-one mappings between these species. The mouse-human homology mapping was obtained from the Mouse Genome Informatics (MGI) database hosted by Jackson Laboratories (*Blake et al., 2017*).

We integrated these three homology maps by representing each map as an undirected bipartite graph, where nodes are genes of one species and edges represent homology between two genes across species. All components were then merged to form an undirected graph where each node represents a gene name and corresponding species. We applied this integrated homology map

consisting of nodes representing the *Drosophila*, mouse, and human dysregulated genes, and all edges induced by the corresponding nodes, to obtain a subgraph consisting of multiple connected components. If any individual connected component contained nodes that belong to all three species, we characterized all genes within the connected component as concordant.

## PPI network and clustering

To examine how the upregulated and downregulated core genes interact functionally, we used STRING v10.5 (*Szklarczyk et al., 2015*). Only high-confidence interactions (edge weight >0.7) were considered. Each node is converted from an ENSEMBL ID to human Entrez ID via the provided mapping file (v10, 04-28-2015). Four subgraphs of STRING were then induced on each core gene set separately. Nodes were further clustered with the InfoMap community detection algorithm (*Rosvall and Bergstrom, 2008*), implemented in the Python iGraph package, with the default settings (trials = 10) (*Csardi and Nepusz, 2000*).

## *Drosophila* behavioral assay

We crossed female virgins that carried the mHTT transgene under the control of either the neuronal or glial driver, or the cell-specific driver alone, to males carrying the experimental allele. We introduced a heat-shock-induced lethality mutation on the Y chromosome ($Y^{P\{hs-hid\}}$) to the disease and cell-specific driver stocks to increase the efficiency of virgin collection (*Starz-Gaiano et al., 2001*). For crosses involving alleles that were lethal or sterile mutations on the X chromosome, this mating strategy was reversed. For behavioral assays, *elav >HTTNT$^{NT231Q128}$* and *repo >HTTNT$^{NT231Q128}$* animals were raised and maintained at 28.5°C. *elav, repo >HTTNT$^{NT231Q128}$* animals were raised and maintained at 25°C. Individual *Drosophila* in each genotype were randomly grouped into replicates of 10.

The negative geotaxis climbing assay was performed using a custom robotic system (SRI International, available in the Automated Behavioral Core at the Dan and Jan Duncan Neurological Research Institute). The robotic instrumentation elicited negative geotaxis by 'tapping' *Drosophila* housed in 96-vial arrays. After three taps, video cameras recorded and tracked the movement of animals at a rate of 30 frames per second for 7.5 s. For each genotype, we collected 4–6 replicates of 10 animals to be tested in parallel (biological replicates). Each trial was repeated three times (technical replicates). The automated, high-throughput system is capable of assaying 16 arrays (1536 total vials) in ~3.5 hr. To transform video recordings into quantifiable data, individual *Drosophila* were treated as an ellipse, and the software deconvoluted the movement of individuals by calculating the angle and distance that each ellipse moves between frames. Replicates were randomly assigned to positions throughout the plate and were blinded to users throughout the duration of experiments. The results of this analysis were used to compute more than two dozen individual and population metrics, including distance, speed, and stumbles.

Software required to run and configure the automation and image/track the videos include Adept desktop, Video Savant, MatLab with Image Processing Toolkit and Statistics Toolkit, RSLogix (Rockwell Automation), and Ultraware (Rockwell Automation). Additional custom-designed software include Assay Control – SRI graphical user interface for controlling the assay machine; Analysis software bundles: FastPhenoTrack (Vision Processing Software), TrackingServer (Data Management Software), ScoringServer (Behavior Scoring Software), and Trackviewer (Visual Tracking Viewing Software).

## In situ and immunofluorescence in HD mouse brain sections

mRNA ISH and immunofluorescence were performed on 25-μm-thick coronal brain sections cut from fresh-frozen brain harvested from a 6-month-old *Hdh$^{zQ175/+}$* mouse. We generated digoxigenin (DIG)-labeled mRNA antisense probes against *Nrxn3* using reverse-transcribed mouse cDNA as a template and an RNA DIG-labeling kit from Roche (Sigma). Primer and probe sequences for the *Nrxn3* probe are available in Allen Brain Atlas (http://www.brain-map.org). ISH was performed by the RNA In Situ Hybridization Core at Baylor College of Medicine using an automated robotic platform as previously described (*Yaylaoglu et al., 2005*) with modifications of the protocol for fluorescent ISH. In brief: after the described washes and blocking steps, the DIG-labeled probe was visualized using a tyramide-Cy3 Plus kit (1:50 dilution, 15 min incubation, Perkin Elmer). Following

washes in phosphate buffered saline (PBS), the slides were stained with 1:500 anti-GFAP rabbit polyclonal antibody (DAKO, Z0334) diluted in 1% blocking reagent in Tris buffered saline (Roche Applied Science, 11096176001) overnight at 4°C. After washing, slides were treated with 1:500 anti-rabbit IgG Alexa 488 secondary antibody for 30 min at room temperature (Invitrogen, A-11008). The slides were stained with DAPI and cover slipped using ProLong Diamond (Invitrogen, P36970). Images were taken at ×63 magnification using a Leica SP8 confocal microscope.

For visualizing Serpina1, 8.5-month-old male zQ175 mice (four wildtype and three knock-in) were deeply anesthetized and transcardially perfused with 1× PBS. The tissues were then treated with 70% ethanol for 24 hr, 95% ethanol overnight, 100% ethanol for 4 hr, and chloroform overnight. Next tissues were treated with paraffin at room temperature overnight and again with paraffin at 65°C for 2 hr. Paraffin-embedded tissue blocks were coronally sectioned at the thickness of 8 μm, starting from Bregma 0.98 mm. Immunofluorescence for Serpina1 was conducted with rabbit anti-Serpina1a primary antibody (Invitrogen, PA5-16661), followed by the biotin labeled secondary antibody and detected by Alexa Fluor 488 conjugated streptavidin. Fluorescent imaging of the striatal region was performed on a Leica Sp8 confocal microscope.

## Immunoblot of *Drosophila* lysates

For all immunoblot experiments, *Drosophila* were raised and maintained at 25°C. Female F1 progeny were collected and flash-frozen 24 hr after eclosion. Heads were separated by genotype and divided into eight individuals per replicate. *Drosophila* heads were lysed and homogenized in 30 μL of lysis buffer (1× NuPage LDS Sample Buffer, 10% beta-mercaptoethanol) and boiled at 100°C for 10 min. Lysates were loaded on a 4–12% gradient Bis-Tri NuPage (Invitrogen) gel and run at a constant voltage of 80 V for an hour and then 120 V for 30 min. For mHTT levels, a 20% methanol transfer buffer was used to transfer proteins at 4°C overnight using a 200 mA current. For mCD8::GFP, proteins were transferred using a 10% methanol buffer for 2 hr at 4°C using a 200 mA current.

Prior to antibody treatment, all membranes were treated with blocking solution (5% non-fat milk in 1× TBST). For primary antibody treatment, all antibodies were diluted in blocking solution. To assess mHTT levels, membranes were then treated with a 1:500 mouse anti-HTT solution (mAb5490, EMD Millipore) overnight. For a loading control, membranes were subsequently treated with a 1:1000 alpha-tubulin antibody (Abcam EP1332Y). 1:1000 Rabbit anti-GFP (ThermoFisher A-11122) was used to assess levels of mCD8::GFP, and 1:1000 anti-lamin C (Hybridoma Bank LC28.26) was used as a loading control. All blots were treated with 1:5000 Goat anti-Mouse (IRDye 800CW Goat anti-Mouse IgG) and Goat anti-Rabbit (RDye 680RD Goat anti-Rabbit IgG) secondary antibodies diluted in blocking solution for 1 hr and imaged using the Odyssey CLx imager (LI-COR Biosciences).

## Knockdown of Serpina1a and HTRF in STHdh$^{Q111/Q7}$ cells

ST*Hdh*$^{Q111/Q7}$ cells were reverse transfected with pooled siRNAs using Lipofectamine 2000 (Life Technologies, cat. no. 11668). Cells were treated with a pool of four small siRNAs per gene with the following sequences (Qiagen 1027280):

- *Htt*
    5′- GAAAUUAAGGUUCUGUUGA-3′
    5′- CCACUCACGCCAACUAUAA-3′
    5′- GAUGAAGGCUUUCGAGUCG-3′
    5′- UAACAUGGCUCAUUGUGAA-3′
- *Serpina1a*
    5′- GAAUAUAACUUGAAGACAC-3′
    5′-GGGCUGACCUCUCCGGAAU-3′
    5′- UGGUAGAUCCCACACAUAA-3′
    5′- GAAAGAUAGCUGAGGCGGU-3′
- *Scramble*

Following siRNA treatment, cell lysis buffer (1× PBS with 1% TrintonX-100% and 1% EDTA-free protease inhibitor; Calbiochem, #539134) was added to each well and the plate was put on ice for 30 min. After incubation, cells were homogenized and lysates were extracted. Separately, HTRF assay buffer was prepared using 50 mM NaH$_2$PO$_4$ (pH 7.4), 400 mM KF, 0.1% bovine serum albumin, 0.05% Tween-20, and Quant-ITTM PicoGreen (1:1500). The donor antibody, 2B7 conjugated to

terbium, was diluted in HTRF assay buffer to a concentration of 0.023 µg/mL, and the acceptor antibody, mAb2166 (SigmaAldrich) conjugated to fluorescent dye D2, was diluted to a final concentration of 1.4 µg/mL. 5 µL of the HTRF buffer was added to 5 µL of cell lysates (5 µL) in each well of a 384-well plate. Lysates were then incubated at 4°C overnight.

HTRF was performed in a Perkin Elmer EnVision multilabel plate reader (model #2104), measuring the 615 nM and 665 nM, as well as the PicoGreen signal at 485 nM. Each sample was measured following 30 cycles of the excitation at an interval of 16.6 ms.

## DEG identification in *Drosophila* HD models

Differential expression analysis used the DESeq2 R package on a total of 12 comparisons (two HD models, two cell-specific drivers, and three timepoints) (*Love et al., 2014*). Outlier detection was performed using PCA on normalized gene expression data, resulting in one sample being removed. To establish a list of upregulated and downregulated DEGs in *Drosophila*, we examined the FDR at every timepoint in both genetic models. If the FDR was <0.05 at any data point in the HD models compared to control, we established that that gene was dysregulated due to the presence of mHTT in either neurons or glia. We did not take the magnitude of fold-change into account, only the direction (upregulated or downregulated) (*Langfelder et al., 2016*).

## Reanalysis of HD patient-derived and knock-in mouse model transcriptomes

The identification of DEGs from humans was based on microarray data from brain tissue collected post-mortem in patients with HD and age-matched, healthy individuals. For consistency with the reported results, we examined the summary statistics of the caudate probe on the Affymetrix U133 A and B microarrays. We computed the FDR by applying the Benjamini–Hochberg procedure to the p-values reported in *Hodges et al., 2006*. A probe was said to be dysregulated if the absolute value of its fold-change was >1.2 (or $\log_2 FC > 0.263$) and the FDR was <0.05. Since multiple Affymetrix probes can match to the same Entrez ID, we specified that an Entrez-identified human gene was dysregulated, if there exists a matching probe that is also dysregulated.

We established the lists of upregulated and downregulated DEGs in mice from RNA-seq data presented in *Langfelder et al., 2016*, where the authors profiled mRNA of an allelic series in a HD knock-in mouse model. We reanalyzed data from the striatum at 6 months, identifying gene expression alterations that were significant (FDR < 0.05) in the continuous-Q case, a summary regression variable derived from DESeq that tests the association of the expression profile with Q-length as a numeric variable (*Love et al., 2014*).

## Connectivity of the mHTT responding networks compared to a striatal proteome background

We randomly sampled 471 proteins (equivalent to the average number of input proteins in the mHTT Responding networks) 1000 times from 15,884 proteins that are expressed in the striatum. Implementing the same parameters that were used for the mHTT responding networks, we constructed clustered PPI networks with the random striatal protein lists as inputs. We calculated the average node degree and average node betweenness within each network of random genes and compiled a distribution using these results. A Z-score was calculated using the distribution compiled from the random striatal networks. These Z-scores were then used to calculate the p-values that are reported in *Supplementary file 2*. All simulations and statistical calculations were performed in R—this script can be found as *Source code 2*.

## Analysis of behavioral screen in *Drosophila*

We assessed behavior in *Drosophila* as the speed at which individual animals within one vial moved as a function of age and genotype using a nonlinear random mixed effects model regression. Specifically, we looked at differences in regression between genotypes with time (additive effect, represented by a shift in the curve) or the interaction of genotype and time (interactive effect, represented by a change in the slope of the curve). We estimated the expected statistical power to detect differences by each of our models using a stringent threshold for statistical significance (alpha = 0.001). We reported p-values representative of the pairwise post-hoc tests for testing

whether all possible pairs of genotype curves are different in both models. We considered differences between positive controls and experimental perturbations of p<0.001 to be significant. p-values were adjusted for multiplicity using Holm's procedure. Code for this analysis is available upon request from the Botas Laboratory. All graphing and statistical analyses were performed in R.

### Statistical analysis for western blot and HTRF

Images of western blots were analyzed using the Image Studio Lite software. We used an equivalent area to measure signal intensity across all replicates. We present proteins of interest as a ratio of the target protein to loading control (n = 5 immunoblots). Experimental replicates were compared to controls using a one-sided Student's t-test. For HTRF, levels of mHTT were calculated by taking the ratio of the fluorescence signals (665 nM/615 nM) and normalizing to the PicoGreen signal in experimental groups after subtracting the signal from wells containing only sample buffer and HTRF buffer, without protein lysates. Results are presented as the average and standard error of the mean of the $\Delta F$ (%) ($\Delta F$ (%) = (Sample ratio − blank ratio)/blank ratio $\times 100$). Each treatment group consisted of nine replicates (n = 9). p-values were calculated using Fisher's LSD test.

### Statistical analysis for immunohistochemistry

The mean intensity for images of Serpina1a stained brain slices was measured using ImageJ. For each sample, five images were measured and the mean was calculated The control group consisted of four samples (n = 4), and the HD group consisted of three samples (n = 3). Groups were compared using a two-tailed t-test assuming unequal variances.

## Acknowledgements

We thank Vicky Brandt for critical input on the manuscript. We also thank Steve Goldman for sharing data cited in this work and his thoughtful insight. This work was supported by grants to JB from NIH/NIA (R01AG057339) and CHDI. BL is sponsored by Natural Science Foundation of China (31970747, 31601105, 81870990, 81925012). TO and MM were supported by the NIGMS Ruth L Kirschstein National Research Service Award (NRSA) Predoctoral Institutional Research Training Grant (T32 GM008307) provided to the Genetics and Genomics Graduate Program at Baylor College of Medicine. AL was supported by Baylor College of Medicine Medical Scientist Training Program and the NLM Training Program in Biomedical Informatics and Data Science (T15 LM007093) at the Gulf Coast Consortium. The High Throughput Behavioral Screening core at the Jan and Dan Duncan Neurological Research Institute was supported by generous philanthropy from the Hildebrand family foundation. The project was also supported by a shared Instrumentation grant from the NIH (S10 OD016167) and Baylor College of Medicine IDDRC Grant Number P50HD103555 from the Eunice Kennedy Shriver National Institute of Child Health and Human Development for use of the Microscopy Core facilities, the Cell and Tissue Pathogenesis Core, and the RNA In Situ Hybridization Core facility with the expert assistance of Dr. Cecilia Ljungberg. The content is solely the responsibility of the authors and does not necessarily represent the official views of the Eunice Kennedy Shriver National Institute of Child Health and Human Development or the National Institutes of Health.

## Additional information

### Funding

| Funder | Grant reference number | Author |
| --- | --- | --- |
| Natural Science Foundation of China | 31970747 | Boxun Lu |
| Natural Science Foundation of China | 31601105 | Boxun Lu |
| Natural Science Foundation of China | 81870990 | Boxun Lu |
| Natural Science Foundation of China | 81925012 | Boxun Lu |

| National Research Centre | T32 GM008307 | Tarik Seref Onur Megan Mair |
| NLM | T15 LM007093 | Andrew Laitman |
| NIH | S10 OD016167 | Juan Botas |
| Eunice Kennedy Shriver National Institute of Child Health and Human Development | P50HD103555 | Juan Botas |
| NIH | R01AG057339 | Juan Botas |
| CHDI Foundation | I-0986 | Juan Botas |

The funders had no role in study design, data collection and interpretation, or the decision to submit the work for publication.

## Author contributions

Tarik Seref Onur, Data curation, Software, Formal analysis, Supervision, Validation, Investigation, Visualization, Methodology, Writing - original draft, Writing - review and editing; Andrew Laitman, Data curation, Software, Formal analysis, Funding acquisition, Validation, Investigation, Visualization, Methodology, Writing - original draft; He Zhao, Ryan Keyho, Hyemin Kim, Jennifer Wang, Huilan Wang, Data curation, Formal analysis, Investigation; Megan Mair, Data curation, Software, Formal analysis, Investigation; Lifang Li, Formal analysis, Investigation; Alma Perez, Investigation, Methodology; Maria de Haro, Resources, Investigation, Methodology; Ying-Wooi Wan, Data curation, Software, Formal analysis, Investigation, Visualization, Methodology; Genevera Allen, Software, Formal analysis, Visualization, Methodology; Boxun Lu, Conceptualization, Supervision, Investigation, Methodology; Ismael Al-Ramahi, Conceptualization, Resources, Investigation, Methodology; Zhandong Liu, Software, Supervision, Investigation, Methodology, Project administration; Juan Botas, Conceptualization, Resources, Supervision, Funding acquisition, Validation, Investigation, Methodology, Writing - original draft, Project administration, Writing - review and editing

## Author ORCIDs

Tarik Seref Onur ⓘ https://orcid.org/0000-0002-3234-6263
Juan Botas ⓘ https://orcid.org/0000-0001-5476-5955

## Decision letter and Author response

Decision letter https://doi.org/10.7554/eLife.64564.sa1
Author response https://doi.org/10.7554/eLife.64564.sa2

# Additional files

## Supplementary files

• Source code 1. R script for prefiltering differentially expressed genes (DEGs) in *Drosophila* Huntington's disease (HD) models.

• Source code 2. R script for identification of differentially expressed genes (DEGs) in *Drosophila* Huntington's disease (HD) models.

• Source code 3. Python code for identification of differentially expressed genes (DEGs) in postmortem human Huntington's disease (HD) striata.

• Source code 4. Python code for identification of differentially expressed genes (DEGs) from Huntington's disease (HD) mouse models.

• Source code 5. Python code for homology mappings across humans, mice, and *Drosophila*.

• Source code 6. Python code for comparing Huntington's disease (HD) differentially expressed genes (DEGs) across species.

• Source code 7. Python code for clustering networks.

• Source code 8. R script for network analysis in a striatal background summarized in ***Supplementary file 2***.

- Source code 9. R script for analyzing *Drosophila* behavioral assays.
- Supplementary file 1. Mappings of orthologous Huntington's disease (HD) human, mouse, and *Drosophila* differentially expressed genes (DEGs).
- Supplementary file 2. Network connectivity of differentially expressed genes (DEGs) responding to glial or neuronal mutant Huntingtin (mHTT) expression.
- Supplementary file 3. Cluster membership and summary of cluster annotations for differentially expressed genes (DEGs) responding to glial or neuronal mutant Huntingtin (mHTT) expression.
- Supplementary file 4. Summary of statistical analysis for alleles in the Synapse Assembly cluster that modify glial mutant Huntingtin (mHTT)-induced behavioral impairments.
- Supplementary file 5. Summary of alleles screened and results for common modifiers of mutant Huntingtin (mHTT)-induced behavioral impairments in neurons and glia.
- Supplementary file 6. Summary of statistics for alleles that are common suppressors of neuronal and glial mutant Huntingtin (mHTT)-induced behavioral impairments.
- Transparent reporting form

## Data availability

RNA-sequencing data produced by this study has been deposited in GEO under accession code GSE157287. We have provided source data for figures 2—6, and for figure 3-figure supplement 1, figure 5-figure supplement 1, and figure 6-figure supplements 1—3. Codes for analyzing gene expression, networks, and *Drosophila* behavior are provided.

The following dataset was generated:

| Author(s) | Year | Dataset title | Dataset URL | Database and Identifier |
|---|---|---|---|---|
| Onur TS, Laitman A, Perez A, Wan YW, Al-Ramahi I, Liu Z, Botas J | 2020 | RNA-sequencing of *Drosophila* expressing mutant Huntingtin in neurons or glia | https://www.ncbi.nlm.nih.gov/geo/query/acc.cgi?acc=GSE157287 | NCBI Gene Expression Omnibus, GSE157287 |

The following previously published datasets were used:

| Author(s) | Year | Dataset title | Dataset URL | Database and Identifier |
|---|---|---|---|---|
| Osipovitch M, Asenjo-Martinez A, Cornwell A, Dhaliwal S, Zou L, Chandler-Militello D, Wang S, Li X, Benraiss S-J, Lampp A, Benraiss A, Windrem M, Goldman SA | 2018 | hESC-based human glial chimeric mice reveal glial differentiation defects in Huntington disease | https://www.ncbi.nlm.nih.gov/geo/query/acc.cgi?acc=GSE105041 | NCBI Gene Expression Omnibus, GSE105041 |
| Langfelder P, Gao F, Wang N, Howland D, Kwak S, Vogt TF, Aaronson JS, Rosinski J, Coppola G, Horvath S, Yang WX | 2016 | Transcriptome profiling in knock-in mouse models of Huntington's disease [striatum; cortex; liver; tissue survey] | https://www.ncbi.nlm.nih.gov/geo/query/acc.cgi?acc=GSE65776 | NCBI Gene Expression Omnibus, GSE65776 |
| Hodges A, Strand AD, Aragaki AK, Kuhn A, Sengstag T, Hughes G, Elliston LA, Hartog C, Goldstein DR, Thu D, Hollingsworth ZR, Collin F, Synek B, Holmans PA, Young | 2006 | Human cerebellum, frontal cortex [BA4, BA9] and caudate nucleus HD tissue experiment | https://www.ncbi.nlm.nih.gov/geo/query/acc.cgi?acc=GSE3790 | NCBI Gene Expression Omnibus, GSE3790 |

AB, Wexler NS,
Delorenzi M,
Kooperberg C,
Augood SJ, Faull
RL, Olson JM,
Jones L, Luthi-
Carter R

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
