## [Decision Letter]

**Acceptance summary:**

This is an exciting study that began with hypotheses about the role of glial cells in Huntington disease from bioinformatic comparisons and modeling from previously published datasets, which were then validated and explored in a *Drosophila* model. The paper is methodologically varied and innovative, and will be of interest both to scientists directly in the Huntington's disease field, and more broadly in the study of neurobiology and other neurological disorders.

**Decision letter after peer review:**

Thank you for submitting your article "Downregulation of glial genes involved in synaptic function mitigates Huntington's Disease pathogenesis" for consideration by *eLife*. Your article has been reviewed by two peer reviewers, and the evaluation has been overseen by Michael Eisen as the Senior Editor and Reviewing Editor, who also acted as a third reviewer. The following individual involved in review of your submission has agreed to reveal their identity: Leslie M Thompson (Reviewer #1).

The reviewers have discussed their reviews with one another and were enthusiastic about this work. The major discovery was the downregulation of synaptic glial genes that mitigate the HD phenotype, and the concept of compensatory versus pathogenic genes. Also the novel finding of a key gene that relates to the finding of synaptic glial gene dysregulation and shows an effect on HTT levels. While the concept that reducing synaptic gene expression is not entirely novel as a strategy, previous studies did not focus on glia and also were primarily focused on calcium associated genes.

The Reviewing Editor has drafted this to help you prepare a revised submission.

Essential Revisions:

As you will see below, the reviewers were very positive about this work. They did a few areas that should be addressed in a revision. None should require additional experimentation – just modifications to the manuscript.

1. A majority of the comparisons against human models were using the human embryonic stem cell-derived glial progenitors and not a more mature state of oligodendrocytes or astrocytes. This may be the reason for the few overlapping DEGs in particular to astrocyte progenitors. This should be discussed, as well as the fact that, by limiting their investigation into CD44+ and CD140+ cells, the authors limited the power of the wide array of conserved functions *Drosophila* glia have with mammalian brain cell types other than astrocytes and oligodendrocytes.

2. It would be helpful for authors to discuss that *Drosophila* do not have documented microglia or the same type of brain endothelial cells (but have glial cells that form the fly equivalent of BBB), which fits in to why author has only examined astrocytes in the immuno studies of Nrxn3.

3. Although all data can be reconstructed from the provided data and scripts, there are a few cases where additional supplemental data would be helpful to readers, such as lists of the DEGs in Figures 1B, 1C and similar places where someone interested in specific sets of DEGs would find a supplemental table useful.

4. The reviewers felt that some additional discussion of the genes in Figure 5B,C. would be helpful.

*Reviewer #1:*

This work provides a mechanistic investigation into the contributions of glia to Huntington's disease pathogenesis. The authors first used transcriptomic comparisons across human, mouse, and *Drosophila* models to generate hypotheses of evolutionarily conserved, disease-relevant genes that were tested in high-throughput behavioral *Drosophila* assays. The major discovery was the downregulation of synaptic glial genes that mitigate the HD phenotype.

• The results are surprising in that further downregulation in glia of synaptic genes are compensatory to HD pathogenesis and offer a new way to think about some of these consistent gene expression changes as well as place within glial context.

Strengths of this paper include:

• The authors took advantage of previously published datasets to demonstrate concordently dysregulated genes across human, mouse, and two *Drosophila* (neuronal and/or glial) HD models. This dissection of neuronal vs glial-driven pathogenesis is a valuable aspect of this study and highlights the importance of an often-understudied population of cells in neurological diseases. The study further highlights the power of mining published data sets and testing specific hypotheses and is a very comprehensive study.

• Their technique of focusing on striatal-enriched proteins for performing protein-protein interaction networks and modeling connectivity between DEGs is appreciated in the context of the disease, bringing more power to their findings.

• The elucidation of compensatory vs pathogenic genes in HD glia is novel.

• Manuscript in present form describes methodological reasoning and detailed discussions behind results identified.

• All RNA-seq and source data used in this manuscript are clearly provided. Code for *Drosophila* analysis is also provided

*Reviewer #2:*

The work of Onur et al. is a timely and elegant unbiased analysis RNAseq of mHTT expressed in neurons compared to glia in fly models of HD. This is integrated with known transcriptomic data sets from HD human and mouse models. Evaluation of a cluster of genes involved in synaptic assembly when mHTT was expressed in glia cells is the focus of the analysis. The major strength of the work is the evaluation of modifiers in HD based on their comprehensive approach. Some sections of the results could have more clarity as to the rationale of the next step in the experiments. The data generated, the modifiers for HD identified will have a significant impact on the field.

---

## [Author Response]

Essential Revisions:As you will see below, the reviewers were very positive about this work. They did a few areas that should be addressed in a revision. None should require additional experimentation – just modifications to the manuscript.1. A majority of the comparisons against human models were using the human embryonic stem cell-derived glial progenitors and not a more mature state of oligodendrocytes or astrocytes. This may be the reason for the few overlapping DEGs in particular to astrocyte progenitors. This should be discussed, as well as the fact that, by limiting their investigation into CD44+ and CD140+ cells, the authors limited the power of the wide array of conserved functions *Drosophila* glia have with mammalian brain cell types other than astrocytes and oligodendrocytes.2. It would be helpful for authors to discuss that *Drosophila* do not have documented microglia or the same type of brain endothelial cells (but have glial cells that form the fly equivalent of BBB), which fits in to why author has only examined astrocytes in the immuno studies of Nrxn3.

Both point 1 and point 2 are now discussed in the first paragraph of the revised Discussion section of the manuscript

3. Although all data can be reconstructed from the provided data and scripts, there are a few cases where additional supplemental data would be helpful to readers, such as lists of the DEGs in Figures 1B, 1C and similar places where someone interested in specific sets of DEGs would find a supplemental table useful.

Lists of all human, mouse and fruit fly concordantly altered (upregulated and downregulated) DEGs in Figures 1B, 1C are included in Figure 1-source data 1 and Supplementary File 1.

Also, supplemental data for Figure 5 including statistical analyses and controls can be found in Supplementary Files 5 and 6 and Figure 5—figure supplements 1B and 1C. All these data are referenced in the corresponding main figures.

4. The reviewers felt that some additional discussion of the genes in Figure 5B,C. would be helpful.

To discuss these genes a whole new paragraph has been added to the section of Results titled: “Reducing SERPINA1 function mitigates behavioral impairments in neurons and glia, and lowers mHTT protein levels”